# MissDAG: Causal Discovery in the Presence of Missing Data with Continuous Additive Noise Models

**Erdun Gao**[1][*][†]   **Ignavier Ng**[2][*]   **Mingming Gong**[1]   **Li Shen**[3]   **Wei Huang**[1]
**Tongliang Liu**[4]   **Kun Zhang**[2,5]   **Howard Bondell**[1]
[1] The University of Melbourne, [2] Carnegie Mellon University,
[3] JD Explore Academy, [4] The University of Sydney
[5] Mohamed bin Zayed University of Artificial Intelligence
erdun.gao@student.unimelb.edu.au, {ignavierng, kunz1}@cmu.edu
{mingming.gong, wei.huang, howard.bondell}@unimelb.edu.au
shenli100@jd.com, tongliang.liu@sydney.edu.au

## Abstract

State-of-the-art causal discovery methods usually assume that the observational data is complete. However, the missing data problem is pervasive in many practical scenarios such as clinical trials, economics, and biology. One straightforward way to address the missing data problem is first to impute the data using off-the-shelf imputation methods and then apply existing causal discovery methods. However, such a two-step method may suffer from suboptimality, as the imputation algorithm may introduce bias for modeling the underlying data distribution. In this paper, we develop a general method, which we call MissDAG, to perform causal discovery from data with incomplete observations. Focusing mainly on the assumptions of ignorable missingness and the identifiable additive noise models (ANMs), MissDAG maximizes the expected likelihood of the visible part of observations under the expectation-maximization (EM) framework. In the E-step, in cases where computing the posterior distributions of parameters in closed-form is not feasible, Monte Carlo EM is leveraged to approximate the likelihood. In the M-step, MissDAG leverages the density transformation to model the noise distributions with simpler and specific formulations by virtue of the ANMs and uses a likelihood-based causal discovery algorithm with directed acyclic graph constraint. We demonstrate the flexibility of MissDAG for incorporating various causal discovery algorithms and its efficacy through extensive simulations and real data experiments.

## 1   Introduction

Discovering the underlying causal relations among variables of interest often occupies a prominent position for supporting stable inference and rational decisions [42] in many applications such as medical diagnostics [48], recommendation systems [64] and economics [27]. To achieve this goal, conducting randomized controlled trials or using interventions is often acknowledged as the golden rule, which is effective but challenging in practice owing to high costs, ethical issues, or difficulties in obtaining compliance [47]. To address this issue, causal discovery from purely observational data, which may be more realistic in specific settings, has drawn considerable attention in both academic and industrial fields [58, 24, 23, 21].

Existing causal discovery methods, such as constraint-based methods [57, 13], score-based methods [12, 46], and methods based on functional causal models [53, 25, 72, 44], typically focus on the

---

[*]Equal contribution.
[†]Work was done during an internship at JD Explore Academy.

36th Conference on Neural Information Processing Systems (NeurIPS 2022).

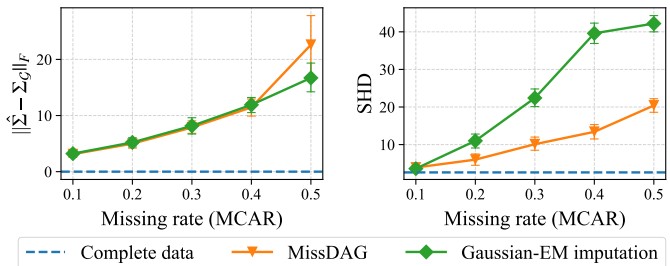

Figure 1: Example of linear Gaussian model with equal noise variance. With zero-mean noises, the recovered covariance matrix $\hat{\Sigma}$ (sufficient statistic) can be a criterion for distribution recovery.

settings in which complete observations are available. However, in practice, datasets often suffer from missing values caused by various factors such as entry errors, deliberate non-responses, and sampling drops [31]. Following the definitions by Little and Rubin [31], the missing types can be categorized into three classes, namely missing completely at random (MCAR), missing at random (MAR), and missing not at random (MNAR), according to different missing mechanisms. Many previous efforts have focused specifically on figuring out more identifiable MNAR cases [6, 34, 37] and estimating the causal graphs from some specific MNAR cases [17, 62], while less attention has been paid to the M(C)AR cases as no extra assumption is required to recover the ground-truth data distribution from incomplete observations [51, 34].

To perform causal discovery in the M(C)AR case, a naive approach to handle the missing values is the listwise deletion method that simply drops the samples with missing value(s) in at least one of the variables. However, this may lead to unsatisfactory performance if the sample size is limited or the missing rate is high [59, 62] because of the decreased statistical power. Another straightforward approach is to impute those missing values instead of listwise deleting them. However, these imputation methods may introduce bias for modeling the underlying data distribution [29]. Moreover, as shown in Fig. (1), even though Gaussian-EM imputation method can consistently recover the data distribution as there is no model misspecification, it may lead to sub-optimal directed acyclic graph (DAG) estimation because it focuses solely on distribution recovery instead of structure learning. Therefore, a principled causal discovery approach that can handle the M(C)AR case is needed.

**Contributions.** In this work, we develop *a practical and general EM-based framework*, called MissDAG, to perform causal discovery in the presence of missing data, in which the underlying missing mechanism is independent from the observed information, which includes the M(C)AR case. Considering the data generating model, we focus on the identifiable additive noise models (ANMs) [44], including the linear non-Gaussian model [53], linear Gaussian model with equal noise variance [45], and nonlinear ANMs [25, 44]; as a byproduct, our framework also accommodates the typical non-identifiable case, namely the linear Gaussian model with non-equal noise variances that can only be identified to Markov equivalence class [57]. The resulting MissDAG framework flexibly accommodates different score based causal discovery algorithms [12, 74, 39, 71] developed for complete data and can be potentially extended to deal with more general cases (e.g., the log-determinant term is incorporated in the likelihood function). Moreover, we conduct extensive experiments on a variety of settings, including synthetic and real data, against many baselines to verify the effectiveness of MissDAG.

## 2 Preliminaries

**Additive noise models.** We adopt the notion of structural causal model (SCM) [42, 43] to characterize the causal relations among variables. Each SCM $\mathcal{M} = \langle \mathcal{Z}, \mathcal{X}, \mathcal{F} \rangle$ consists of the exogenous variable set $\mathcal{Z} = \{Z_1, Z_2, \ldots, Z_d\}$, the endogenous variable set $\mathcal{X} = \{X_1, X_2, \ldots, X_d\}$, and the function set $\mathcal{F} = \{f_1, f_2, \ldots, f_d\}$. Here, each function $f_i$ computes the variable $X_i$ from its parents (or causes) $\mathbf{Pa}_{X_i}$ and an exogenous variable $Z_i$, i.e., $X_i = f_i(\mathbf{Pa}_{X_i}, Z_i)$. In this work, we focus on a specific class of SCMs, called the ANMs [25, 44], given by

$$X_i = f_i(\mathbf{Pa}_{X_i}) + Z_i, \quad i = 1, 2, \ldots, d, \tag{1}$$

where $Z_i$, interpreted as the additive noise variable, is assumed to be independent with variables in $\mathbf{Pa}_{X_i}$ and mutually independent with variables in $\mathcal{Z} \backslash Z_i$.

**Causal graph.** Each SCM $\mathcal{M}$ induces a causal graph, which we assume in this work to be a DAG[3]. The DAG $\mathcal{G}_{\mathcal{M}} = (\mathbb{V}, \mathbb{E})$ consists of a vertex set $\mathbb{V} := \{1, 2, \ldots, d\}$, in which each node $i$ corresponds to the variable $X_i$, and an edge set $\mathbb{E} \subseteq \mathbb{V}^2$ where $(i, j) \in \mathbb{E}$ if and only if $X_i \in \mathbf{Pa}_{X_j}$. Let $X = (X_1, X_2, \ldots, X_d)$ be a random vector that includes the variables in $\mathcal{X}$, and $P(X)$ (with density $p(x)$) be the joint distribution of random vector $X$. We assume that there are no latent common causes of the observed variables (i.e., *causal sufficiency*), which, together with the *acyclicity* assumption, indicates that $P(X)$ and induced DAG $\mathcal{G}_{\mathcal{M}}$ satisfy the causal Markov condition [42].

# 3  Problem definition

**Notations.** In this paper, we focus on the finite-sample setting with missing data. Consider a dataset $\mathcal{D} = (\mathbf{X}, \mathbf{Y})$ that consists of $N$ samples, where $\mathbf{X} \in \mathbb{R}^{N \times d}$ and $\mathbf{Y} \in \mathbb{R}^{\widetilde{N} \times d}$. Each row $(\mathbf{X}_i, \mathbf{Y}_i)$, independently sampled from $P(X, Y)$, represents the $i$-th observation. $\mathbf{Y}$ is the indicator matrix that records the missing positions in $\mathbf{X}$, i.e., $\mathbf{Y}_{ij} = 0$ if $\mathbf{X}_{ij}$ is missing and $\mathbf{Y}_{ij} = 1$ otherwise. In the presence of missing data, the fully observed data $\mathbf{X}$ is unavailable. For simplicity, let $\mathbf{O}$ group all observed positions in $\mathbf{X}$ and $\mathbf{M}$ group all the missing positions. For each observation $\mathbf{X}_i$, let $\mathbf{o}$ group the indexes of the observed part and $\mathbf{m}$ group the indexes of the missing part of $\mathbf{X}_i$. Notice that $\mathbf{o}$ and $\mathbf{m}$ are different for different observations. Then, $\mathbf{X_O} = [\mathbf{X}_{ij} : \mathbf{Y}_{ij} = 1, i \in [N], j \in [d]]$ includes all observed positions and $\mathbf{X_M} = [\mathbf{X}_{ij} : \mathbf{Y}_{ij} = 0, i \in [N], j \in [d]]$[4] includes all missing positions in $\mathbf{X}$. Similarly, we have $\mathbf{X}_{i\mathbf{o}} = [\mathbf{X}_{ij} : \mathbf{Y}_{ij} = 1, j \in [d]]$ and $\mathbf{X}_{i\mathbf{m}} = [\mathbf{X}_{ij} : \mathbf{Y}_{ij} = 0, j \in [d]]$.

Following [31], we define the full likelihood of the $i$-th sample $(\mathbf{X}_{i\mathbf{o}}, \mathbf{Y}_i)$ as

$$\mathcal{L}_{\text{full}}(\mathbf{X}_{i\mathbf{o}}, \mathbf{Y}_i; \theta, \psi) = \int p(\mathbf{X}_{i\mathbf{o}}, \mathbf{X}_{i\mathbf{m}}; \theta) \, p(\mathbf{Y}_i | \mathbf{X}_{i\mathbf{o}}; \psi) \, \mathrm{d}\mathbf{X}_{i\mathbf{m}},$$

$$= p(\mathbf{Y}_i | \mathbf{X}_{i\mathbf{o}}; \psi) \int p(\mathbf{X}_{i\mathbf{o}}, \mathbf{X}_{i\mathbf{m}}; \theta) \, \mathrm{d}\mathbf{X}_{i\mathbf{m}},$$

where the parameters $\psi$ govern the missing mechanisms and $\theta$ include all the model parameters. The ignorable likelihood of $\mathbf{X}_{i\mathbf{o}}$ is defined as

$$\mathcal{L}_{\text{ignorable}}(\mathbf{X}_{i\mathbf{o}}; \theta) = \int p(\mathbf{X}_{i\mathbf{o}}, \mathbf{X}_{i\mathbf{m}}; \theta) \, \mathrm{d}\mathbf{X}_{i\mathbf{m}}.$$

**Assumption 1** (Ignorable missingness [31]). *The inference about parameters $\theta$ based on the ignorable likelihood evaluated only by $\mathbf{X}_{i\mathbf{o}}$ is the same as inference for $\theta$ based on the full likelihood.*

Ignorable missingness is important as it a common assumption that is required by EM-style algorithms and also our method. It can be interpreted as a belief that the available data is sufficient to "correct" the missing data, by assuming that the missingness and model parameters are distinct.

**Task definition.** Consider a distribution $P(X, Y)$, where the marginal distribution $P(X)$ is induced from a SCM $\mathcal{M}$ satisfying the assumption of ANM as defined in Eq. (1), and a dataset $\mathcal{D} = (\mathbf{X}, \mathbf{Y})$. In practice, the observed distribution is only $P(X_o, Y)$ satisfying Assumption 1 and the observational part of the dataset is $\widetilde{\mathcal{D}} = (\mathbf{X_O}, \mathbf{Y})$. Our task is to learn the DAG $\mathcal{G}_{\mathcal{M}}$ from the dataset $\widetilde{\mathcal{D}}$.

# 4  MissDAG

In this section, we introduce our proposed method, called MissDAG, which leverages the penalized EM framework to iteratively identify the causal graph $\mathcal{G}_{\mathcal{M}}$ and model parameters from the incomplete data $\widetilde{\mathcal{D}}$. In the M-step, MissDAG takes the log-likelihood of the observational part of the sample and applies a penalty function as the score function to guide the search of model parameters. Instead of directly modeling the complex likelihood of the sample, MissDAG equivalently models the simpler noise distributions by the density transformation [35] since ANMs always limit the noise distributions to some specific distribution families. Moreover, the prior information of causal structure is also considered as an inductive bias to reduce the variance of parameter estimation. In the E-step, the

---

[3]Indeed, there may be difference between a DAG and a causal graph–the directed edges of the latter is given a causal meaning that allows it to answer interventional queries [28].

[4]We use $[N] = \{1, 2, \ldots, N\}$ to represent the set of all integers from 1 to $N$.

log-likelihood function is integrated over the posterior of missing entries to obtain the expectation if the closed-form of the posterior is available. Otherwise, Monte Carlo (MC) simulations [65] are adopted to numerically compute the expectation. The details are shown in the following subsections.

## 4.1 The overall framework of MissDAG

Leveraging the development of score based causal discovery methods, we follow the style that taking the log-likelihood of observations (only the observational part) and a penalty function as the score function. Then, the general form of the optimization problem can be written as

$$\arg\max_{\theta} \ \mathcal{S}(\theta) = \log p(\mathbf{X_O}; \theta) - \lambda \, \mathbf{PEN}(\theta),$$

$$\text{subject to } \theta_{\mathcal{G}} \in \mathbf{DAGs},$$

where $\theta = (\theta_{\mathcal{G}}, \theta_{\mathcal{F}})$, including a graph learning part $\theta_{\mathcal{G}}$ and a causal mechanisms learning part $\theta_{\mathcal{F}}$, denotes the parameters of an SCM $\mathcal{M}$. However, in some methods [74, 75, 39], $\theta_{\mathcal{G}}$ can be absorbed into and induced from $\theta_{\mathcal{F}}$. $\mathbf{PEN}(\cdot)$ is the penalty function and $\lambda$ is the penalty coefficient. With the i.i.d. assumption of each observation $(\mathbf{X}_i, \mathbf{Y}_i)$, the score function $\mathcal{S}(\theta)$ can be written as

$$\mathcal{S}(\theta) = \sum_{i=1}^{N} \log \int p(\mathbf{X}_{i\mathbf{o}}, \mathbf{X}_{i\mathbf{m}}; \theta) \, \mathrm{d}\mathbf{X}_{i\mathbf{m}} - \lambda \, \mathbf{PEN}(\theta). \tag{2}$$

Unfortunately, the closed-form solution of $\mathcal{S}(\theta)$ cannot be obtained. Since the score function in Eq. (4.1) gives rise to a penalized maximum log-likelihood estimation problem, we can take the iterative penalized EM method [11], which relates the parameters estimation of the SCM from $\log p(\mathbf{X_O}; \theta)$ to the same parameters estimation from the complete-data log-likelihood $\log p(\mathbf{X}; \theta)$.

Different from the imputation methods that replace the missing entries $\mathbf{X_M}$ by some specific values, EM based methods formulate a two-step iterative operation. We start with an initial value $\theta^0$ and denote $\theta^t$ as the estimate of $\theta$ at the $t$-th iteration. Then, each iteration of the EM method can be represented as the following two steps:

- **E-step** takes the estimated model parameters of the previous step and the observational part $\mathbf{X_O}$ to impute the missing entries by the distribution of $\mathbf{X_M}$, which, in other words, gets the expected log-likelihood of the complete-data as follows.

$$\mathcal{Q}(\theta, \theta^t) = \int p(\mathbf{X_M}|\mathbf{X_O}; \theta^t) \log p(\mathbf{X_O}, \mathbf{X_M}; \theta) \, \mathrm{d}\mathbf{X_M} - \lambda \, \mathbf{PEN}(\theta)$$
$$= \mathbb{E}_{X_m|X_o; \theta^t} \{\log p(\mathbf{X_O}, \mathbf{X_M}; \theta)\} - \lambda \, \mathbf{PEN}(\theta). \tag{3}$$

- **M-step** calculates $\theta^{t+1}$ by maximizing the $\mathcal{Q}$ function as follows.

$$\theta^{t+1} \in \arg\max_{\theta} \mathcal{Q}(\theta, \theta^t)$$

The E-step calculates the expected data log-likelihood $\mathcal{Q}(\theta, \theta^t)$. The following M-step, then, maximizes $\mathcal{Q}(\theta, \theta^t)$ in $\theta$ for the fixed $\theta^t$ with DAG constraint. The convergence analysis of MissDAG is provided in Appendix E.

### 4.1.1 Log-likelihood term of $\mathcal{S}(\theta)$

The final problem comes to the exact formulation of $p(\mathbf{X}; \theta)$, which is not straightforward to obtain in our problem since ANMs typically impose assumptions on the noise distribution instead of the joint distribution $P(X)$. Benefiting from the well-researched results of the change of variables rule of density transformation [35], we can equivalently formulate $p(\mathbf{X}; \theta)$ by transforming it to $p(\mathbf{Z}; \theta)$. Here, for simplicity, we take $Z = (Z_1, Z_2, \ldots, Z_d)$ and $f = (f_1, f_2, \ldots, f_d)$. Then, we have

$$p_X(X) = p_Z(X - f(X))|\det(\mathbf{I} - \mathbf{J}_f)|, \tag{4}$$

where $\mathbf{I}$ is the identity matrix and $\mathbf{J}_f$ represents the Jacobian of $f$ evaluated on $X$.

**Proposition 1.** *If $\theta_{\mathcal{G}}$ represents a DAG, then $|\det(\mathbf{I} - \mathbf{J}_f)| = 1$.*

The proof is included in Appendix B.1. With Proposition 1, the log-determinant term becomes zero if the candidate solution is acyclic and can be dropped to simplify Eq. (5). However, we would also like to point out that some recent score-based structure learning methods based on continuous optimization, e.g., GOLEM [39] and NOTEARS-ICA [73], have shown that including the log-determinant term (which corresponds to likelihood based on directed cyclic graphs) in its objective function may be desirable and lead to better performance. In this work, this term is ignored since only acyclic models are considered. Then, with Eq. (4) and Proposition 1, we can model the log-likelihood of the simpler mutually independent noises distributions. Then, we have

$$\mathcal{L}(\mathbf{X}_i; \theta) = \sum_{j=1}^{d} \log p_{Z_j}(\mathbf{X}_{ij} - f_j(\mathbf{X}_i)) + \log |\det(\mathbf{I} - \mathbf{J}_f)| = \sum_{j=1}^{d} \log p_{Z_j}(\mathbf{X}_{ij} - f_j(\mathbf{X}_i)). \quad (5)$$

In the E-step, with different data generation models, the exact formulation of the posterior may be unavailable. In the next section, therefore, we split up these two cases named exact posterior and approximate posterior respectively for presentations.

## 5  Different posterior cases

### 5.1  Exact posterior

Firstly, we deal with the linear Gaussian models including linear Gaussian model with equal variance (LGM-EV) and linear Gaussian model with non-equal variance (LGM-NV). That is to say, in Eq. (1), each $f_i \in \mathcal{F}$ is a linear function and $Z_i \sim \mathcal{N}(0, \sigma_i^2)$ with $\Sigma_Z = \text{diag}(\sigma_1^2, \sigma_2^2, \dots, \sigma_d^2)$. Then, the model can be rewritten as

$$X = W^T X + Z, \quad (6)$$

where $W \in \mathbb{R}^{d \times d}$ is the weight matrix and $W_{ij} \neq 0$ means that $X_i$ is one of the causes of $X_j$. Based on the density of a linear transformation, we know $P(X)$ belongs to a multivariate Gaussian distribution. For multivariate Gaussian, sufficient statistics consist of the mean vector $\mu(X)$ (the first-order moment) and the covariance matrix $\text{cov}(X)$ (the second-order moment). With zero-mean assumption of $Z$, we have $\mu(X) = \mathbf{0}$ and estimate $\text{cov}(X)$ by $\mathbf{T}$. In other words, we can equivalently replace $p(\mathbf{X}; \theta)$ by $p(\mathbf{T}; \theta)$. Specifically, with full data, $\mathbf{T}$ can be directly calculated by $\mathbf{T} = \frac{1}{N}\mathbf{X}^T\mathbf{X}$.

In this case, we specify $\theta = (W, \Sigma_Z)$ since the two parameters can govern the distribution $P(X)$. According to Eq. (5), the complete log-likelihood $\mathcal{L}(\mathbf{X}; W, \Sigma_Z)$ can be sufficiently formulated by

$$\begin{aligned}
\mathcal{L}(\mathbf{X}; W, \Sigma_Z) &= \log p(\mathbf{T}; W, \Sigma_Z) \\
&= -\frac{1}{2}\text{Tr}(\log \Sigma_Z) - \frac{1}{2N}\text{Tr}((\mathbf{I} - W)^T \mathbf{T}(\mathbf{I} - W)\Sigma_Z) - \frac{d}{2}\log 2\pi. \quad (7)
\end{aligned}$$

Since (7) is linear in $\mathbf{T}$, the $\mathcal{Q}$ function can be formulated in a closed-form.

### 5.1.1  E-step: compute $\mathcal{Q}(W, \Sigma_Z, W^t, \Sigma_Z^t)$

As discussed before, the E-step calculates the expected log-likelihood $\log p(\mathbf{T}; W, \Sigma_Z)$ with $\mathbf{X}_\mathbf{O}$ and $(W^t, \Sigma_Z^t)$. From Eq. (6), we have $X = (\mathbf{I} - W^T)^{-1}Z$. Then, the implicit parameter $\Sigma_X^t$ can be estimated by $\Sigma_X^t = (\mathbf{I} - W)^{-T}\Sigma_Z^t(\mathbf{I} - W)^{-1}$. Then, $\mathbf{T}^t = \mathbb{E}[\mathbf{T} \mid \mathbf{X}_\mathbf{O}; \Sigma_X^t]$ can be straightforwardly calculated by the well-known results on the conditional distributions of the multivariate Gaussian. Each entry of $\mathbf{T}^t$ is obtained by $\mathbf{T}_{ij}^t = [\frac{1}{N}\sum_{k=1}^{N}\xi_k \mid \mathbf{X}_\mathbf{O}; \Sigma_X^t]$, where

$$\xi_k = \begin{cases} \Sigma_{X\,ij}^t + \hat{\mathbf{X}}_{ki}\hat{\mathbf{X}}_{kj} & \text{if } \mathbf{X}_{ki} \& \mathbf{X}_{kj} \text{ are missing,} \\ \hat{\mathbf{X}}_{ki}\hat{\mathbf{X}}_{kj} & \text{otherwise.} \end{cases}$$

$\hat{\mathbf{X}}_k$ records the expectation of missing part of the $k$-th instance $\mathbf{X}_k$ and also shares the same indexes with $\mathbf{X}_k$. $\hat{\mathbf{X}}_k$ is initialized as $\mathbf{X}_k$. Then, $\hat{\mathbf{X}}_{k\mathbf{m}} = \Sigma_{X\,\mathbf{mo}}^t \Sigma_{X\,\mathbf{o}^{-1}\mathbf{o}}^t \mathbf{X}_{k\mathbf{o}}$.

### 5.1.2 M-step: maximize $\mathcal{Q}(W, \Sigma_Z, W^t, \Sigma_Z^t)$

With $\mathbf{T}^t$, the M-step maximizes the score $\mathcal{S}(\theta)$ with $\mathcal{L}(\mathbf{T}^t; W, \Sigma_Z) = \log p(\mathbf{T}^t; W, \Sigma_Z)$. Here, we also plug in the DAG constraint for optimization to ensure that the estimated graph is acyclic. Then, the overall optimization problem can be formulated as

$$W^{t+1}, \Sigma_Z^{t+1} = \underset{W, \Sigma_Z}{\arg\max} \ \mathcal{L}(\mathbf{T}^t; W, \Sigma_Z) + \lambda \ \mathbf{PEN}(W),$$

$$\text{subject to } \mathcal{G}_W \in \mathbf{DAGs},$$

where $\mathcal{G}_W$ means the graph induced from $W$. Here, we do not restrict the use of any specific algorithm for solving this problem. For the LGM-EV, one can adopt the greedy search method by Peters and Bühlmann [45], GOLEM [39], or NOTEARS [74] to estimate the DAG $\mathcal{G}$, while for the LGM-NV, one can apply different search methods like GES [12], A* [71], and GOLEM [39].

## 5.2 Approximate posterior

Unfortunately, for Non-Linear (NL)-ANMs and Linear Non-GAussian Model (LiNGAM) cases, the E-step is not available in a closed-form, which would make the likelihood inference for missing data more difficult. There are mainly two problems to compute the expectation: (1) computing the posterior distribution $p(\mathbf{X}_{im}|\mathbf{X}_{io}; \theta^t)$; (2) computing the integral. Since the problem mainly comes from the noise modeling as we later show, we prefer to first introduce the M-step of our method.

### 5.2.1 M-step: maximize $\mathcal{Q}(\theta, \theta^t)$

From Eq. (5), we know that two reasons lead to the non-closed-form for integral. The first one is that $f$ includes some complex non-linear functions. However, considering the likelihood issue, the non-linearity problem can be well handled by taking neural networks to model $f$ [75, 40, 30]. Therefore we leave $f$ for both linear and non-linear models for brevity. The second problem is non-Gaussian noise, which needs to be clarified before the M-step.

**The problem of $p_Z(Z)$.** If noises are non-Gaussian, the exact formulation of each noise distribution is unknown. To make the likelihood-based methods work, a fixed Super (Sub)-Gaussian prior distribution can be set to model the noise distributions [26, 75]. The theoretical result that the maximum likelihood estimate is locally consistent even in the presence of small misspecification error is well-established [2]. Here, we take the Super-Gaussian distribution as an example:

$$p_i(Z_i) = c_Z \exp\left(-2\log\cosh\left(Z_i\right)\right), \tag{8}$$

where $c_Z$ is a constant. Notice that using (8) needs to assume the unit scale of noise, which may not hold for the real scenarios. Therefore, we prefer to take a normalized likelihood for standardized noise variable $Z_i/\sigma_i$. Then, the standardized log-likelihood[5] of the observation $\mathbf{X}_i$ will be

$$\mathcal{L}_{\text{standardized}}(\mathbf{X}_i; \theta) = \sum_{j=1}^{d} \log p_{Z_j}\left(\frac{\mathbf{X}_{ij} - f_j(\mathbf{X}_i)}{\hat{\sigma}_j}\right) - \log \hat{\sigma}_j, \tag{9}$$

where $\sigma_j^2 = \text{Var}(Z_j)$ is the variance of noise $Z_j$ with empirical version of $\hat{\sigma}_j^2 = \frac{1}{N}\sum_{j=1}^{N}(\mathbf{X}_{:,j} - f_j(\mathbf{X}))^2$. Then, Eq. (9) can serve as the log-likelihood term and the overall optimization problem in the M-step can be written as

$$\theta^{t+1} = \underset{\theta}{\arg\max} \ \mathcal{L}_{\text{standardized}}(\mathbf{X}^t; \theta) - \lambda \ \mathbf{PEN}(\theta),$$

$$\text{subject to } \theta_{\mathcal{G}} \in \mathbf{DAGs},$$

where $\mathcal{L}_{\text{standardized}}(\mathbf{X}^t; \theta)$ represents the expected log-likelihood function in the $t$-th iteration.

### 5.2.2 E-step: compute $\mathcal{Q}(\theta, \theta^t)$

**The problem of the integral.** For LiNGAM and NL-ANMs, there is no closed-form for the integral to get the expectation of log-likelihood in the E-step. Naturally, we take the Monte Carlo sampling

---

[5]The term "standardized log-likelihood" is taken to follow the literature of independent component analysis.

method [50] to approximate the expectation. Then, Eq. (3) can be reformulated as

$$\mathcal{Q}(\theta, \theta^t) = \mathbb{E}_{X_m|X_o;\theta^t}\left\{\log p(\mathbf{X_O}, \mathbf{X_M}; \theta)\right\} = \sum_{i=1}^{N}\sum_{j=1}^{N_s} \log p(\mathbf{X}_{i\mathbf{o}}, x_{i\mathbf{m}}^j; \theta),$$

where $x_{i\mathbf{m}}^j$, sampled from the posterior, represents the $j$-th value of the total $N_s$ sampling results for the missing part $\mathbf{X}_{i m}$ of the observation $\mathbf{X}_i$.

**The problem of sampling from the posterior.** Instead of directly sampling from $p(\mathbf{X}_{i\mathbf{m}}|\mathbf{X}_{i\mathbf{o}}; \theta^t)$ to fill the missing part $\mathbf{X}_{i\mathbf{m}}$, we use rejection sampling [8] to sample from a proposal distribution $Q(X_\mathbf{m})$ with the probability density function $q(X_\mathbf{m})$, from which we can readily draw samples. Then, a constant $c_k$ is set to guarantee that $c_k q(X_\mathbf{m}) \geq p(\mathbf{X}_{i\mathbf{m}}|\mathbf{X}_{i\mathbf{o}}; \theta^t)$ for $\forall X_\mathbf{m}$. For each sample $x_\mathbf{m}^j$ from $q(X_\mathbf{m})$, the accept rate would be $p_{\text{accept}} = p(x_\mathbf{m}^j|\mathbf{X}_{i\mathbf{o}}; \theta^t)/c_k q(x_\mathbf{m}^j)$. However, $p(\mathbf{X}_{i\mathbf{m}}|\mathbf{X}_{i\mathbf{o}}; \theta^t)$ can not be directly obtained. With Bayes Rule, we have

$$p(\mathbf{X}_{i\mathbf{m}}|\mathbf{X}_{i\mathbf{o}}; \theta^t) = \frac{p(\mathbf{X}_i; \theta^t)}{p(\mathbf{X}_{i\mathbf{o}}; \theta^t)},$$

while for each instance $\mathbf{X}_i$, $p(\mathbf{X}_{i\mathbf{o}}; \theta^t)$ is a constant marked as $c_o$. However, the joint distribution $p(\mathbf{X}_i; \theta^t)$ is still not directly available. With Eq. (4) and Proposition 1, we find that we may skip for obtaining the closed-form of $p(\mathbf{X}_i; \theta^t)$ but equivalently provide the value of $p_Z(\mathbf{Z}_i; \theta^t)$:

$$p(\mathbf{X}_i; \theta^t) = |\det(\mathbf{I} - \mathbf{J}_{\theta_f^t})| p_Z(\mathbf{Z}_i; \theta^t) = \prod_{j=1}^{d} p_{Z_j}(\mathbf{X}_{ij} - \theta_{f_j}^t(\mathbf{X}_i)).$$

Then, just like in the M-step, we take a normalized distribution to model the noise distributions. Then, the probability that a sample can be accepted is given by

$$p_{\text{accept}} = \frac{\prod_{j=1}^{d} p_{Z_j}(\mathbf{X}_{ij} - \theta_{f_j}^t(\mathbf{X}_i))}{c_o c_k q(x_\mathbf{m}^j)}.$$

Here, we take $c_r = c_o c_k$ as a re-normalized constant to activate the rejection sampling methods.

## 6 Experiments

We report the empirical results to verify the effectiveness of MissDAG on both synthetic and a biological dataset.

**Baselines.** We mainly take imputation methods as baselines including Mean Imputation, MissForest Imputation [60], and Optimal Transport (OT)-imputation [36] to impute the incomplete data at first,[6] and then apply the causal discovery methods including GOLEM [39], NOTEARS [74], the algorithm ('Ghoshal') by Ghoshal and Honorio [20], NOTEARS-MLP [75], and NOTEARS-ICA [73] to estimate the causal graph for different assumed models (see Appendix D for details.). For LGM, we also include the structural EM method for multivariate Gaussian distribution, which we called Gaussian-EM imputation, to recover the complete data. For LGM-NV that aims to identify the Complete Partial DAG (CPDAG), we also include different searching methods such as A* [71] and GES [12] to solve the optimization problem of the M-step. Also, we include the Test-wise Deletion PC (TD-PC) as a baseline for LGM-NV. For LiNGAM, we also use ICA-LiNGAM and Direct-LiNGAM as baseline methods and put the results in Appendix F.7.2. The detailed implementations of the imputation and structure learning methods, as well as the hyper-parameters of the proposed method, are presented in Appendix C.

**Metrics.** We report the widely used criterion named Structural Hamming Distance (SHD), which refers to the smallest number of edge additions, deletions, and reversals required to transform the recovered DAG into the true one, averaged over 10 random repetitions to assess how the edges differ between the estimated and ground-truth DAG in the identifiable cases. For the non-identifiable case such as LGM-NV, we report the SHD-CPDAG to measure the distances between different CPDAGs. Other criteria such as F1 and recall are included for the supplementary experiments in Appendix F.

---

[6]Results on more imputation methods including GAIN [68], KNNImputer [3], MICE [66] are shown in Appendix F.1.

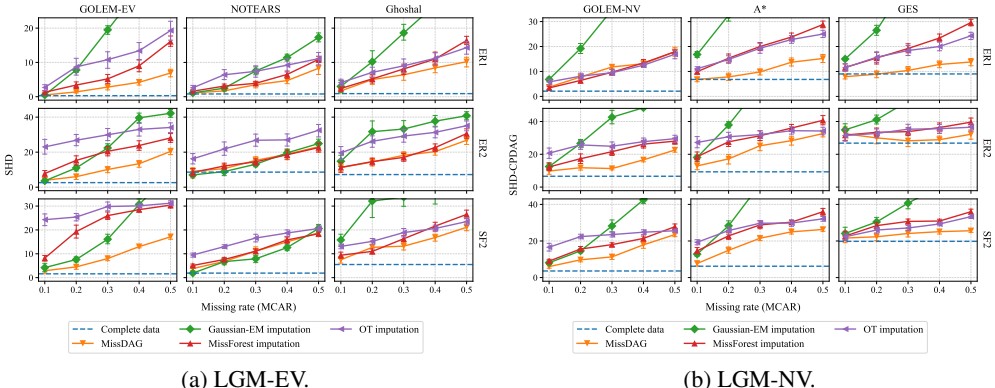

Figure 2: Recovery of the true structure measured by SHD or SHD-CPDAG ($\downarrow$). (a) LGM-EV with $d = 20$. (b) LGM-NV with $d = 15$ since the searching time of A* is too long. Rows: ER1, ER2, and SF2 graphs. Columns: different methods. Some results for Gaussian-EM imputation are truncated because its SHDs are too large in those cases.

**Simulations.** The synthetic data we consider here is generated according to the ANM in Eq. (1). As illustrated above, we consider four cases, including LGM-EV, LGM-NV, LiNGAM, and NL-ANM. In each experiment, a ground-truth DAG $\mathcal{G}$ with $d$ nodes and $kd$ directed edges was first generated from one of the two graph models, Erdős-Rényi (ER) or Scale-Free (SF). According to different edges, $kd$ edges ($k = 1, 2$), the graph model is named ER$k$ or SF$k$. We also simulate denser graphs in Appendix F.6. Then, for linear models, a weighted matrix $W \in \mathbb{R}^{d \times d}$ with coefficients sampled from Uniform($[-2.0, -0.5] \cup [0.5, 2.0]$) with equal probability is generated to assign values to each edge in $\mathcal{G}$. For the non-linear model, corresponding to each edge in $\mathcal{G}$, a $f_i$ is constructed from a fixed MLP with random coefficients. The non-Gaussian noise we take here follows a Gumbel distribution. In line with the settings outlined in [73], we do not consider the scenario of non-equal scales, as the normalized likelihood makes the optimization problem hard to be solved. Our framework, however, can be extended with future advanced methods to tackle this challenge. Experimental results on more different non-linear functions and different noise distributions are included in Appendix F.8 and Appendix F.7.1, respectively. In equal variance/scale sets, all independent noises belongs to their distributions with variance/scale as 1 while non-equal variance/scale settings get the scale of each noise independently sampled from Uniform[1.0, 2.0]. For each experiment, we sample 100 observations for linear models and 200 observations for non-linear models. More results on different numbers of samples with fixed number of nodes and different numbers of nodes with the fixed number of samples are shown in Appendix F.3 and Appendix F.4, respectively. We also add the experiment with $d$ nodes and $2d$ observations with run-time comparisons in Appendix F.5. The missing type in our experiments is MCAR while the results on MAR and MNAR are also provided in Appendix F.2.

**Linear Gaussian case.** In Fig. (2a), across all settings for LGM-EV, including different graphs and missing rates, MissDAG with GOLEM, NOTEARS and Ghoshal as baseline methods can show consistently the best performance or performance comparable to the best performances. While all imputation methods are sensitive to different baseline methods, all NOTEARS-based methods show improvements compared to GOLEM-based methods although GOLEM is the real full likelihood method for LGM. However, GOLEM solves the problem by soft-constraint, which may suffer from the finite sample. For imputation methods, we can see that MissForest usually acquires the second-best place. Gaussian-EM imputation can consistently recover the multivariate Gaussian distribution, but its performance varies a lot with different causal discovery methods. With NOTEARS, Gaussian-EM imputation can gain better, even the best results as compared to the others. The results of LGM-NV are shown in Fig. (2b). Three different searching strategies are considered. It is observed that MissDAG can still achieve the best performances across all settings. The capacities of A* and GES may be limited by the finite observations while GES appears to perform the worst. Moreover, A*, an exact search method, may suffer from high computing complexity. The comparisons on running time are provided in Appendix G.

**Visualization of the learned DAG of MissDAG.** We take an example of the MissDAG optimization process on LGM-EV and plot the change in estimated parameters in Fig. (3), which shows that the

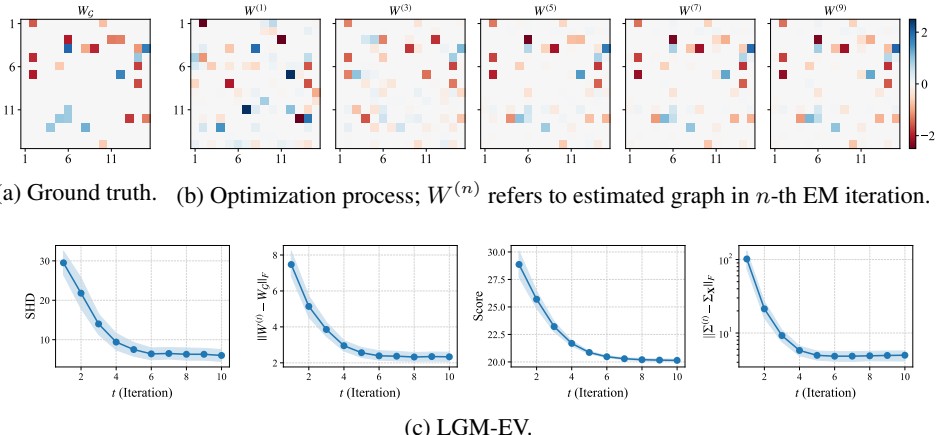

(a) Ground truth.   (b) Optimization process; $W^{(n)}$ refers to estimated graph in $n$-th EM iteration.

(c) LGM-EV.

Figure 3: Visualizations of the optimization process.

learned causal graph asymptotically approximates the ground-truth DAG $\mathcal{G}$, including the existence of edges and their weights. The data distribution can also be well recovered.

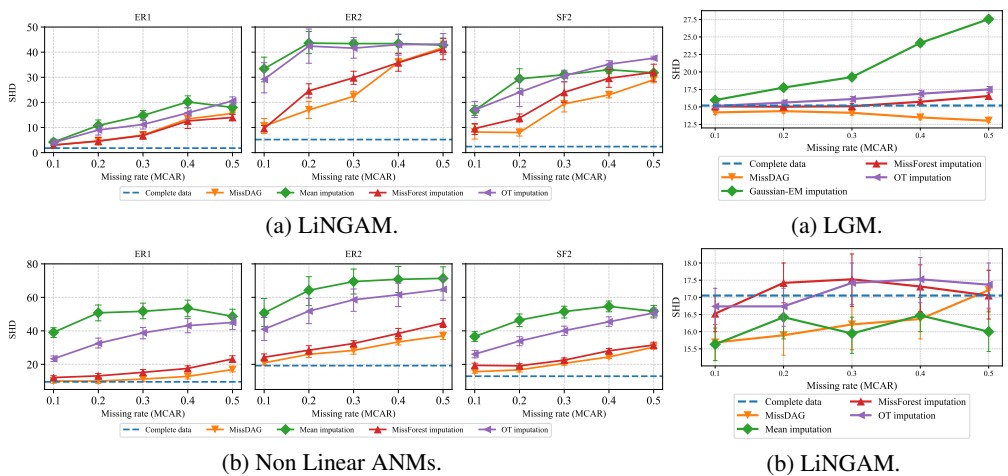

(a) LiNGAM.                                 (a) LGM.

(b) Non Linear ANMs.                       (b) LiNGAM.

Figure 4: MissDAG with approximate posterior.        Figure 5: Dream4 results.

**LiNGAM and NL-ANMs.** Fig. (4a) shows the performances of MissDAG and different baseline methods with NOTEARS-ICA as the causal discovery algorithm on LiNGAM while Fig. (4b) provides the results on NL-ANM of 20 variables. One can see that MissDAG always occupies the best or one of the best methods across all settings. MissForest [60] always shows the best or the second-best performance. Especially for NL-ANM, MissForest can get comparable results to MissDAG.

**Biological dataset.** We take a biological dataset named Dream4 and provided in [22], which simulates gene expression measurements from five sub-networks of transcriptional regulatory networks of E. coli and S. cerevisiae. Here we consider the 10-node networks, which however include feedback loops. We can see that MissDAG occupies the first place for most of the settings and LGM is more suitable to learn from this dataset. The results in Fig. (5) show that there are misspecifications between our models and real data, probably due to the cycles in the real data.

## 7   Related works

**Causal discovery from complete data.** Two lines of methods prevail in causal discovery research, namely constraint-based methods, such as PC and fast causal inference (FCI) [58], and score based methods like GES [12]. The first branch reads the (conditional) independencies information encoded in the data distribution, which can also be viewed as an equality constraint [55], to decide the existence

of edges and directions of some edges. However, the constraint-based method can only reach the Markov Equivalence Class (MEC) of the ground-truth DAG since DAGs in the same MEC share the totally same conditional independencies information. The second branch searches the model parameters in the DAG space by maximizing the penalized likelihood (score) on the observational data. For a long time, these methods suffer from the high searching complexities of combinatorial optimization. Recently, NOTEARS [74] recasts this problem as a continuous optimization by introducing an algebraic characterization of DAG. Then, NOTEARS has been extended to handle non-linear cases [69, 40, 76, 30, 63], time-series data [41], unmeasured confounder [7] and interventional data [9]. By imposing further assumption on the data generating model, NL-ANMs [25, 44], Post-Nonlinear Model (PNL) [72], LGM-EV [45], and LiNGAM [53], etc, are proposed to learn the ground-truth DAG with identifiability guarantees.

**Causality with incomplete data.** MissDeepCausal [33] leverages the deep latent model to estimate the causal effects of a treatment, intervention or policy from incomplete data. GINA [32] systematically analyzes the identifiability of generative models under MNAR case and designs a practical deep generative model which can provide identifiability guarantees for certain MNAR mechanisms. DECI [18] proposes a general deep latent model to perform both causal discovery and inference. Moreover, the theoretical results can guarantee that this model can identify the causal graph under standard causal discovery assumptions.

**BN learning with incomplete data.** Previous methods [23, 56, 15, 16, 49] mainly inherit the Expectation-Maximization (EM) method framework [51], which conducts likelihood inference in an iterative optimization way. Friedman [15], Singh [56] iteratively refine the conditional distributions and sampling the missing values from these distributions. [61] proposed a data augmentation method by a stochastic simulation-based method that draws the filled-in value from a predictive distribution. The augmentation method was accelerated by recasting the problem into two phases, parent set identification by an exact search and structure optimization by an approximate algorithm [1]. However, existing BN learning methods from incomplete data focus on identifying the Markov equivalence classes (i.e., discrete cases) under suitable assumptions and usually formulate the structure learning problem as a discrete optimization program, while our work focuses on continuous identifiable ANMs of which the structure is fully identifiable and includes recent structure learning approaches based on continuous optimization. More discussions can be found in Appendix A.

## 8 Conclusion and Future Work

In this paper, we propose a new approach named MissDAG to learn the underlying causal relations from incomplete data. MissDAG, leveraging the EM-based paradigm, iteratively maximizes the likelihood of the observational part of data with the inductive bias of DAG structure. Existing score-based causal discovery methods can be directly integrated into our framework for graph and model parameter learning. Moreover, MCEM is introduced to address the challenge that the closed-form posterior of missing entries is unavailable. The experiments show that our method works well across various of settings. However, our method inherits the time inefficiency issue of EM algorithm. Future works include (1) improving the sampling efficiency with more efficient sampling or variational inference techniques to approximate the posterior to scale up to larger problems, (2) incorporating other advanced causal discovery methods into the MissDAG framework, and (3) allowing unobserved confounders and cycles, e.g., using the methods by Bhattacharya et al. [7], Ghassami et al. [19].

## Acknowledgements

LS is supported by the Major Science and Technology Innovation 2030 "Brain Science and Brain-like Research" key project (No. 2021ZD0201405). IN and KZ were partially supported by the National Institutes of Health (NIH) under Contract R01HL159805, by the NSF-Convergence Accelerator Track-D award #2134901, by a grant from Apple Inc., and by a grant from KDDI Research Inc.. EG is supported by an Australian Government Research Training Program (RTP) Scholarship. This research was undertaken using the LIEF HPC-GPGPU Facility hosted at the University of Melbourne. This Facility was established with the assistance of LIEF Grant LE170100200. MG was supported by ARC DE210101624. TL was partially supported by Australian Research Council Projects DP180103424, DE-190101473, IC-190100031, DP-220102121, and FT-220100318. HB was supported by ARC FT190100374.

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
