# Supplementary materials

## A  Additional related works

**Comparison with Friedman [15], Singh [56].**  Our work shares similarities with [15, 55], since both rely on the EM algorithm. However, our work focuses on continuous identifiable ANMs that have recently received considerable attention [44], while Friedman [15], Singh [56] focus on discrete cases in which one is only able to identify the Markov equivalence class; therefore, the key technical development is different. (1) For the linear Gaussian case, we derive the closed-form solution of exact posterior that is different from the discrete case considered by Friedman [15], Singh [56]. (2) For the linear non-Gaussian and nonlinear cases, since the exact posterior is not available in closed form, we develop a method based on approximate posterior using Monte Carlo and rejection sampling; such a setup that involves approximate posterior may be more challenging and has not been considered by Friedman [15], Singh [56], since the exact posterior of discrete case considered by Friedman [15], Singh [56] is available in closed form. (3) Our formulation includes modern structure learning approaches based on continuous optimization (in addition to classical methods based on discrete optimization considered by Friedman [15], Singh [56]).

## B  Proofs

### B.1  Proof of Proposition 1

We take $B_{\theta_{\mathcal{G}}}$ as the adjacency matrix of $\theta_{\mathcal{G}}$. If $(B_{\theta_{\mathcal{G}}})_{ij} = 0$, then, $(\mathbf{J}_{\theta_f^t})_{ij} = 0$, i.e., $\mathbf{J}_{\theta_f^t}$ implicitly encodes a DAG structure. Therefore, there exists a permutation matrix $U$ such that $U\mathbf{J}_{\theta_f^t}U^T$ is strictly upper triangular. Then, $|\det(\mathbf{I} - \mathbf{J}_{\theta_f^t})| = |\det(\mathbf{I} - U\mathbf{J}_{\theta_f^t}U^T)| = 1$.

## C  Implementation details

We provide the implementation details for the structure learning and imputation methods, and for the procedure used to generate the missing data. We also describe the hyperparameters used for the proposed MissDAG framework.

### C.1  Structure learning methods

We use existing implementations for most structure learning methods:

- **A\* and GES**[7]: A\* [71, 70] formulates the score-based structure learning problem as a shortest path problem and uses the A\* search procedure with a consistent heuristic function to guide the search in the search space of DAGs, and is guaranteed to return the optimal DAG. On the other hand, GES [12] adopts a greedy search procedure in the search space of equivalence classes. Therefore, in the M-step, one has to convert the estimated equivalence class by GES into a consistent DAG. For both methods, we adopt the BIC score [52].

- **Testwise Deletion PC (TD-PC)**: TD-PC [62] is an extension of PC that makes use of all instances without any missing value for the variables involved in the conditional independence test. It provides asymptotically correct results for the MCAR case while may not give correct for the MAR case, since the condition $p(R|X_m, X_o) = p(R)$ does not hold. Here we use the Fisher-z test and set the $p$-value to 0.05.

- **NOTEARS, NOTEARS-ICA, and NOTEARS-MLP**[8]: NOTEARS-based methods are widely used in our paper, including NOTEARS [74] for the LGM, NOTEAES-ICA (NOTEARS-ICA-MCEM) [73] for the LiNGAM, and NOTEARS-MLP (NOTEARS-MLP-MCEM) [75] for the NL-ANMs. For all NOTEARS-based models, we follow the original papers and use the augmented Lagrangian method to solve the constrained optimization problem; see Appendix D.1 for further

---

[7] https://github.com/cmu-phil/causal-learn
[8] https://github.com/xunzheng/notears

details. We adopt the same hyperparameters suggested in the original papers, since we do not aim to report the best performance for all settings by carefully tuning these parameters. In particular, the initial $\alpha$ and $\rho$ are both set to 0. The other hyperparameters $\gamma$, $h_{min}$ and $\rho_{max}$ are set to $0.25$, $1 \times 10^{-8}$ and $1 \times 10^{16}$, respectively. The sparsity parameter $\lambda_1$ for NOTEARS, NOTEARS-ICA (NOTEARS-ICA-MCEM), NOTEARS-MLP (NOTEARS-MLP-MCEM) is set to $0.1$, $0.1$, and $0.03$, respectively. Moreover, NOTEARS-MLP (NOTEARS-MLP-MCEM) also applies a $\ell_2$ penalty to all the weights of the multilayer perceptrons, whose coefficient $\lambda_2$ is set to $0.01$.

- **GOLEM**[9]: Ng et al. [39] show that, when likelihood-based objective is used together with the soft sparsity and DAG constraints, it is able to to recover the true structure under certain conditions. They further proposed an algorithm, called GOLEM, to do so, which involves solving an unconstrained optimization problem. The hyperparameters need for GOLEM is just (1) $\lambda_1$ for the $\ell_1$ sparsity and (2) $\lambda_2$ for the DAG constraint penalty. Throughout all experiments, we set them to $\lambda_1 = 5 \times 10^{-2}$ and $\lambda_2 = 5 \times 10^{-3}$, which are slightly larger than the original ones used by Ng et al. [39] and found to be more effective when the sample size is small. In the NOTEARS-ICA-MCEM and MOTEARS-MLP-MCEM, to construct $q(X_{\mathbf{m}})$, we first use zero imputation to impute $\mathbf{X_O}$ and get $\hat{\mathbf{X}}$. For each observation $\mathbf{X}_i$ and $j \in \mathbf{m}$, we calculate $\sigma_j^2$ from $\hat{\mathbf{X}}$. Then, a diag matrix is constructed as $\Sigma = \mathrm{diag}(\sigma_j^2 : j \in \mathbf{m})$ to be a covariance matrix. Finally, $q(X_{\mathbf{m}})$ is defined as a multivariate Gaussian distribution with zero-mean and $\Sigma$ as the covariance matrix.

- **ICA-LiNGAM and Direct-LiNGAM**[10]: ICA-LiNGAM [53] utilizes independent component analysis to estimate the LiNGAM, while Direct-LiNGAM [54] recovers the causal order of the variables by iteratively removing the effect of each variable from the data.

- **Ghoshal**: The algorithm described by Ghoshal and Honorio [20] first estimates the inverse covariance matrix, and then iteratively identifies and removes a terminal node. The parent set and edge weights are also estimated during the iterative procedure. We adopt our own implementation of the algorithm because we did not manage to find a publicly available implementation. The original algorithm employs the CLIME method [10] to estimate the inverse covariance matrix, while we use the graphical Lasso method [14].

Note that the experiments for GOLEM are conducted on NVIDIA V100 GPU, while those for the other methods are conducted on CPU instances.

### C.2 Imputation methods

We use existing implementations for most imputation methods:

- **MissForest imputation and KNNImputer**[11]: The default hyperparameters are used.

- **MICE imputation**[12]: We set the hyperparameter *n-imputations* as 1.

- **GAIN and optimal transport (OT) imputation**[13]: GAIN is an adversarial learning framework that consists of two generators, which are used to impute the missing entries and generate the hint matrix, respectively, and of a discriminator that is used to distinguish between observed and imputed entries. The hyperparameter $\alpha$ is set to 10 and the hint rate is set to $0.9$. The learning rate is taken as $1 \times 10^{-3}$ and there are a total of 10000 iterations for the adversarial learning procedure. OT imputation leverages the optimal transport distances and integrate it into the loss functions to achieve the imputation. In OT imputation, we set the learning rate and $\epsilon$ to $0.01$, the number of iterations to $500$, and the scaling parameter in Sinkhorn iterations to $0.9$.

- **Mean and Gaussian-EM imputation**: We adopt our own implementation of these two imputation methods. Mean imputation fills the missing entries using the average of the observed values of the corresponding variable. For Gaussian-EM imputation, the E-step is the same as our method, while in the M-step, the estimated statistic $\mathbf{T}$ is directly return to the E-step without any further operation. There is no extra hyperparameter used for these two method.

---

[9]https://github.com/ignavierng/golem
[10]https://github.com/cdt15/lingam
[11]https://github.com/epsilon-machine/missingpy
[12]https://github.com/scikit-learn/scikit-learn
[13]https://github.com/trentkyono/MIRACLE

### C.3 Missingness

According to the underlying reasons why the data are missing, the missing mechanisms are typically classified into three categories. (1) MCAR. The missing mechanism is independent from all variables. (2) MAR. The missing mechanism is systematically related to the observational variables but independent from the missing variables. (3) MNAR. The missing mechanism is related to the missing variables. We describe the procedure to generate the missing matrix $\mathbf{Y}$, which is used to mask the synthetic data $\mathbf{X}$ to simulate different types of missing data, i.e., MCAR, MAR, and MNAR with missing rate $r_m$.

- **MCAR**: Firstly, sampling a matrix $\mathbf{Y}'$ from a $\mathrm{Uniform}([0,1])$, and set $\mathbf{Y}_{ij} = 0$ if $\mathbf{Y}'_{ij} \leq r_m$ and $\mathbf{Y}_{ij} = 1$ otherwise. For the experiment in Appendix F.2 that compare against different missing types, we specifically set $30\%$ of the variables to be full-observational, i.e., without any missing value, in order to ensure a relatively fair comparison with the MAR case.

- **MAR**: We set $30\%$ of the variables to be fully-observed. Then, the missingness of the remaining variables are generated according to a logistic model with random weights that are related to the fully observed variables.

- **MNAR**: The self-masked missingness is taken as the MNAR mechanism. To ensure a relatively fair comparison with the MAR case, $30\%$ of the variables do not have any missing value. Then, the remaining variables are masked according to a logistic model with random weights that are related to the corresponding variables.

### C.4 Hyperparameters of MissDAG

The proposed MissDAG framework is able to solve four types of ANMs, including LGM-EV, LGM-NV, LiNGAM, and NL-ANM. For different models, different causal discovery or structure learning methods are leveraged, each of which involves a different set of hyperparameters, described in Appendix C.1. To ensure a fair comparison, we use the same set of hyperparameters for these structure learning methods across different imputation methods, and our MissDAG framework. Our framework involves an additional hyperparameter corresponding to the number of iterations for the EM procedure, which we set to 10.

## D  Solving the optimization problem

### D.1  Augmented Lagrangian method

Here, we rewrite the equality constrained optimization problem in the M-step (here, we equivalently minimize the negative score function.) of MissDAG as follows:

$$\arg\min_{\theta} \ -\mathcal{S}(\theta) + \lambda\,\mathbf{PEN}(\theta),$$

$$\text{subject to } \theta_{\mathcal{G}} \in \mathbf{DAGs}, \iff h(\theta_{\mathcal{G}}) = \mathrm{Tr}(\exp(\theta_{\mathcal{G}})) - d = 0,$$

where $\exp(\cdot)$ is the matrix exponential and $\mathrm{Tr}(\cdot)$ calculates the matrix trace. In NOTEARS [74], the above optimization problem is solved by leveraging the augmented Lagrangian method [4, 5] to get an approximate solution. It is an iterative-based optimization method, which transforms the optimization object into a series of unconstrained sub-problems. The $t$-th sub-problem involving the augmented Lagrangian can be formulated as

$$\arg\min_{\theta} \ -\mathcal{S}(\theta) + \lambda\,\mathbf{PEN}(\theta) + \alpha_t h(\theta_{\mathcal{G}}) + \frac{\rho_t}{2}|h(\theta_{\mathcal{G}})|^2,$$

where $\alpha_t$ and $\rho_t$ are the parameters updated by the iterative step, which represent the estimate of the Lagrange Multiplier and the penalty parameter, respectively. The values of these two parameters are gradually increased to make the final solution approximately meet the requirement of equality constraint. Specifically, the iterative step follows the following update rules:

$$\theta^{t+1} = \arg\min_{\theta} \ -\mathcal{S}(\theta) + \lambda\,\mathbf{PEN}(\theta) + \alpha_t h(\theta_{\mathcal{G}}) + \frac{\rho_t}{2}|h(\theta_{\mathcal{G}})|^2,$$

$$\alpha^{t+1} = \alpha^t + \rho^t h(\theta_{\mathcal{G}}^t)$$

$$\rho^{t+1} = \begin{cases} \beta\rho^t & \text{if } h(\theta_{\mathcal{G}}^{t+1}) \geq \gamma h(\theta_{\mathcal{G}}^t), \\ \rho^t & \text{otherwise,} \end{cases}$$

where $\beta$ and $\gamma$ are the hyperparameters. For NOTEARS-ICA and NOTEARS-MLP, we also use the augmented Lagrangian method to solve the problem in which the only difference is the score function.

## D.2 Soft constraints

GOLEM [39] employs likelihood-based objective with soft sparsity and DAG constraints for structure learning. However, in our setting of missing data, NOTEARS fits better into our MissDAG framework, specifically in the M-step, as compared to GOLEM, as the former solves a constrained optimization problem and is guaranteed to return DAGs. However, this does not lead to the conclusion that the least squares loss used by NOTEARS is better than the likelihood-based objective used by GOLEM. This is because the study by Ng et al. [39] shows that under certain conditions, likelihood-based objective with the soft constraints introduced are able to to recover the true structure. In other words, unlike NOTEARS, we do not have to enforce a hard acyclicity constraint, and the unconstrained optimization problem will return a solution close to being a DAG in practice. The optimization problem is as follows,

$$\underset{\theta_{\mathcal{M}}}{\arg\min} -\mathcal{S}_{\text{GOLEM}} + \lambda_1 \, \mathbf{PEN}(\theta_{\mathcal{M}}) + \lambda_2 h(\theta_{\mathcal{G}}).$$

where $\lambda_1$ and $\lambda_2$ are the penalty coefficients. With the likelihood-based objective, GOLEM can also be applied to the non-identifiable LGM-NV case to identify the Markov equivalence class of the ground-truth DAG.

## D.3 Thresholding

As suggested by Zheng [73], Ng et al. [39], the solutions produced by the methods described in Appendices D.1 and D.2 usually contain a number of entries with a small magnitude; therefore, thresholding is used to alleviate this problem. For these methods, we follow the original papers and set the threshold to $0.3$ to prune the learned adjacency matrix to get the final graph. To guarantee the DAG output, an iterative deletion method is also taken, which cut off the edge with the minimum magnitude until obtaining a DAG. Note that we only apply this post-processing step after the EM procedure ends to obtain the final graph, but not during the EM procedure.

# E Convergence analysis

The convergence of MissDAG is highly relied on the convergence properties of EM [67] and MCEM framework [38]. Moreover, the convergence analysis of Bayesian network learning from incomplete data has been well provided by Städler and Bühlmann [59], Friedman [16]. For completeness, we provide similar conclusions in this section that meet the different cases in our framework.

To prove the convergence of MissDAG, a penalized EM-based iterative method, we can turn to prove the establishment of $p(X_o; \theta^{t+1}) \geq p(X_o; \theta^t)$ or, equally, $\log p(X_o; \theta^{t+1}) \geq \log p(X_o; \theta^t)$. With Bayes Rule, we have

$$\log p(X_o; \theta) = \log p(X_o, X_m; \theta) - \log p(X_m | X_o; \theta).$$

And then, we get the expectation over the missing variables given the observed variables and the parameters $\theta_t$ on both sides of the equation to obtain

$$\mathbb{E}_{X_m | X_o; \theta^t} \{\log p(X_o; \theta)\} = \mathbb{E}_{X_m | X_o; \theta^t} \{\log p(X_o, X_m; \theta)\} - \mathbb{E}_{X_m | X_o; \theta^t} \{\log p(X_m | X_o; \theta)\}$$
$$\Rightarrow \log p(X_o; \theta) = \mathbb{E}_{X_m | X_o; \theta^t} \{\log p(X_o, X_m; \theta)\} - \mathbb{E}_{X_m | X_o; \theta^t} \{\log p(X_m | X_o; \theta)\}.$$

The first term of the RHS with sparsity constraint corresponds to the term $\mathcal{Q}(\theta, \theta^t)$ in our paper. Since $\theta^{t+1} = \arg\max_\theta \mathcal{Q}(\theta, \theta^t)$, we have

$$\mathcal{Q}(\theta^{t+1}, \theta^t) \geq \mathcal{Q}(\theta^t, \theta^t). \tag{10}$$

Furthermore, to prove $\log p(X_o; \theta^{t+1}) \geq \log p(X_o; \theta^t)$, we also need

$$\mathbb{E}_{X_m | X_o; \theta^t} \{\log P(X_m | X_o; \theta^{t+1})\} \leq \mathbb{E}_{X_m | X_o; \theta^t} \{\log P(X_m | X_o; \theta^t)\},$$

or, equivalently,

$$\mathbb{E}_{X_m|X_o;\theta^t} \left\{ \log \frac{p(X_m|X_o;\theta^{t+1})}{p(X_m|X_o;\theta^t)} \right\} \leq 0.$$

This is straightforward to be proved since

$$\mathbb{E}_{X_m|X_o;\theta^t} \left\{ \log \frac{p(X_m|X_o;\theta^{t+1})}{p(X_m|X_o;\theta^t)} \right\} = -D_{KL}\left(p(X_m|X_o;\theta^{t+1}), p(X_m|X_o;\theta^t)\right)$$
$$\leq 0.$$

To ensure the increase of log-likelihood in each EM iteration, we need Eq. (10) to hold. Using exact search methods, which can search the total parameter space, this can be guaranteed since $\theta^{t+1} = \arg\max_\theta \mathcal{Q}(\theta, \theta^t)$ can be guaranteed. Then, similar to the previous conclusions by Wu [67], Städler and Bühlmann [59], Friedman [16], MissDAG can also reach the stationary points of the overall optimization problem. However, if we use gradient-based methods (e.g., NOTEARS) to solve the optimization, this inequality can not always hold since the DAG constraint is non-convex. That is to say, we cannot guarantee to find better $\theta^{t+1}$. Even though the convergence property does not hold in this case, experimental results provided in our paper still demonstrate the effectiveness of our MissDAG.

# F  Supplementary experiments

## F.1  More baselines

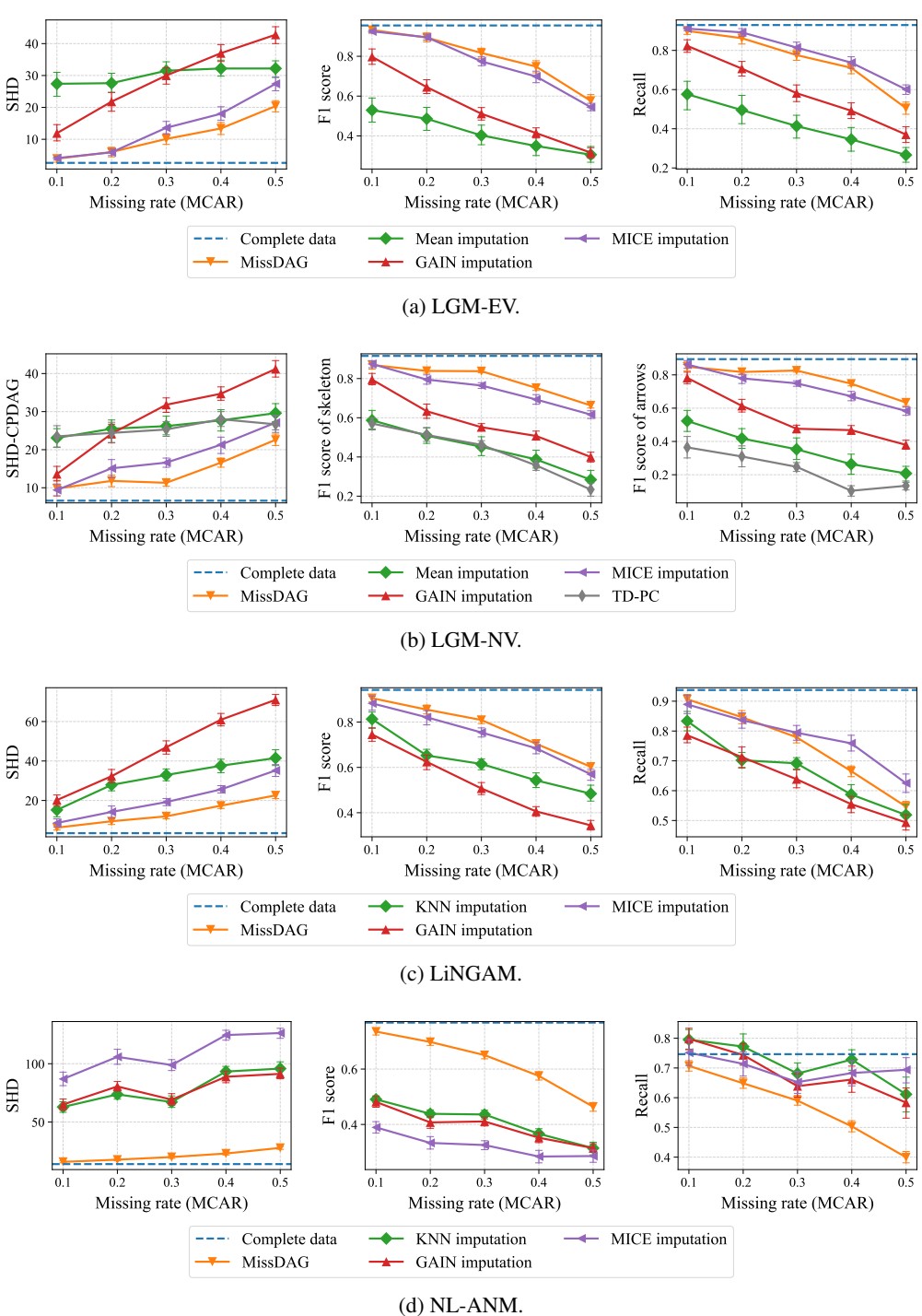

(a) LGM-EV.

(b) LGM-NV.

(c) LiNGAM.

(d) NL-ANM.

Figure 6: Results of comparisons to more baseline methods.

## F.2    Different missing types

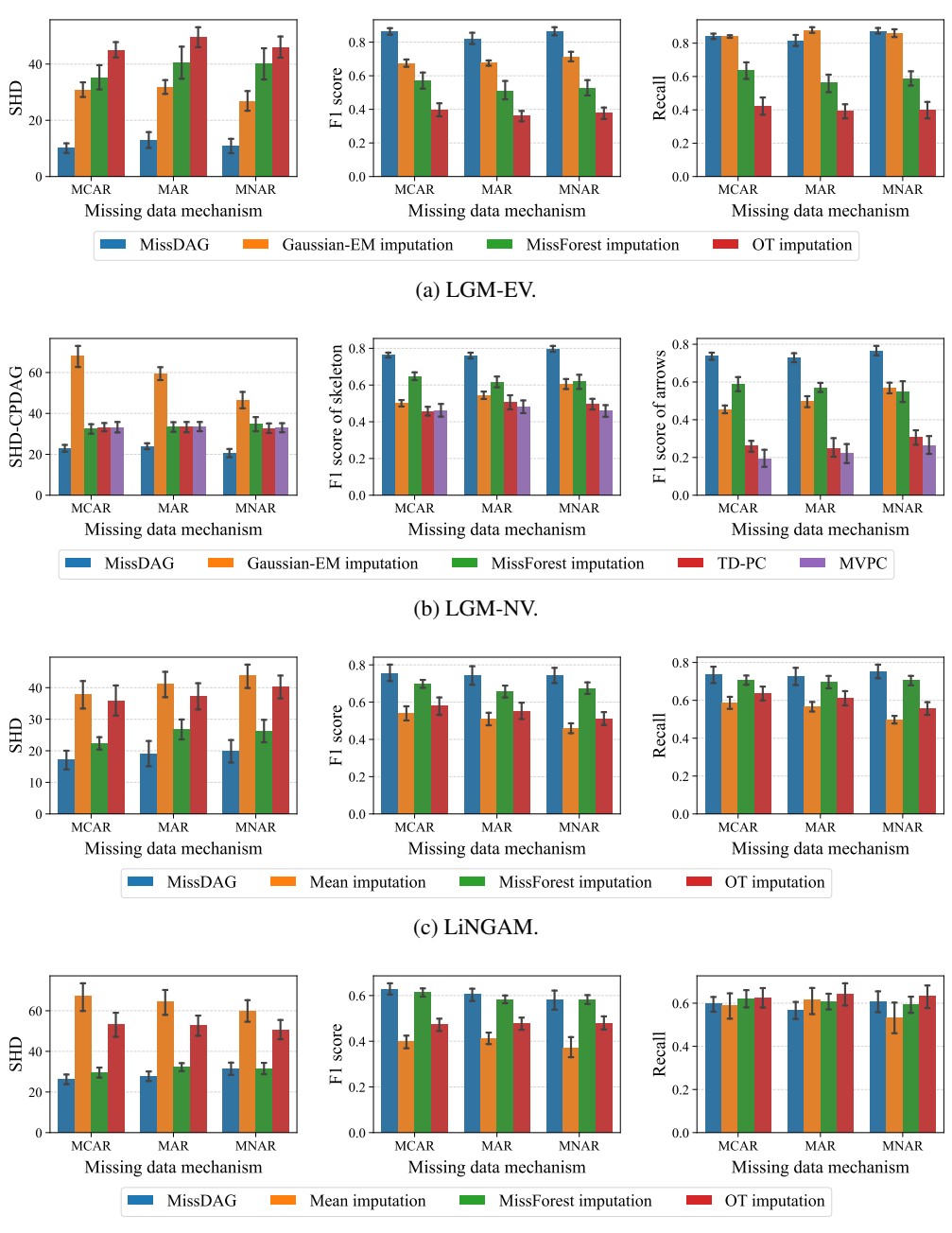

(a) LGM-EV.

(b) LGM-NV.

(c) LiNGAM.

(d) NL-ANM.

Figure 7: Results on different missing mechanisms.

## F.3 Different numbers of samples

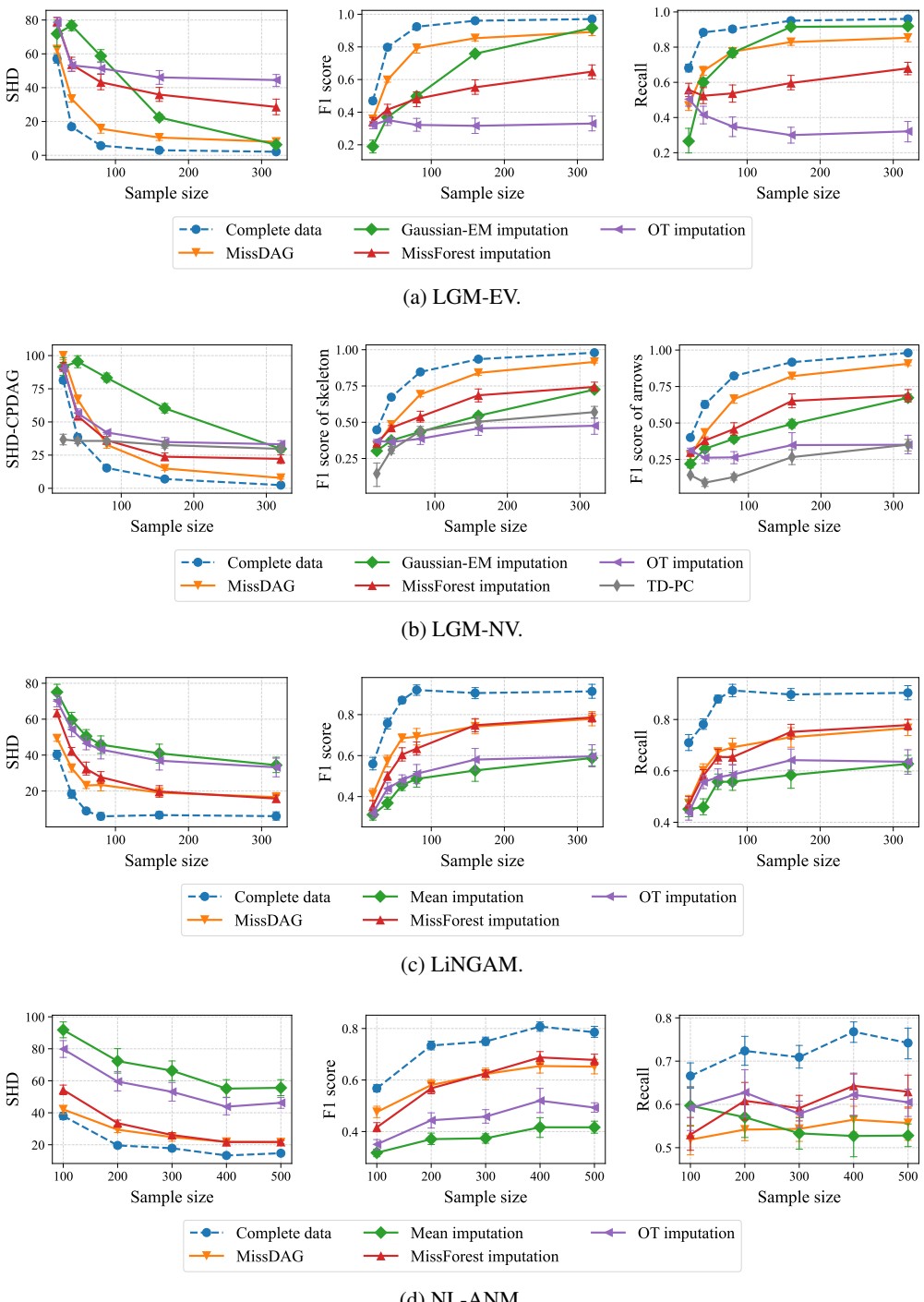

(a) LGM-EV.

(b) LGM-NV.

(c) LiNGAM.

(d) NL-ANM.

Figure 8: Results on different number of samples.

## F.4 Different numbers of nodes

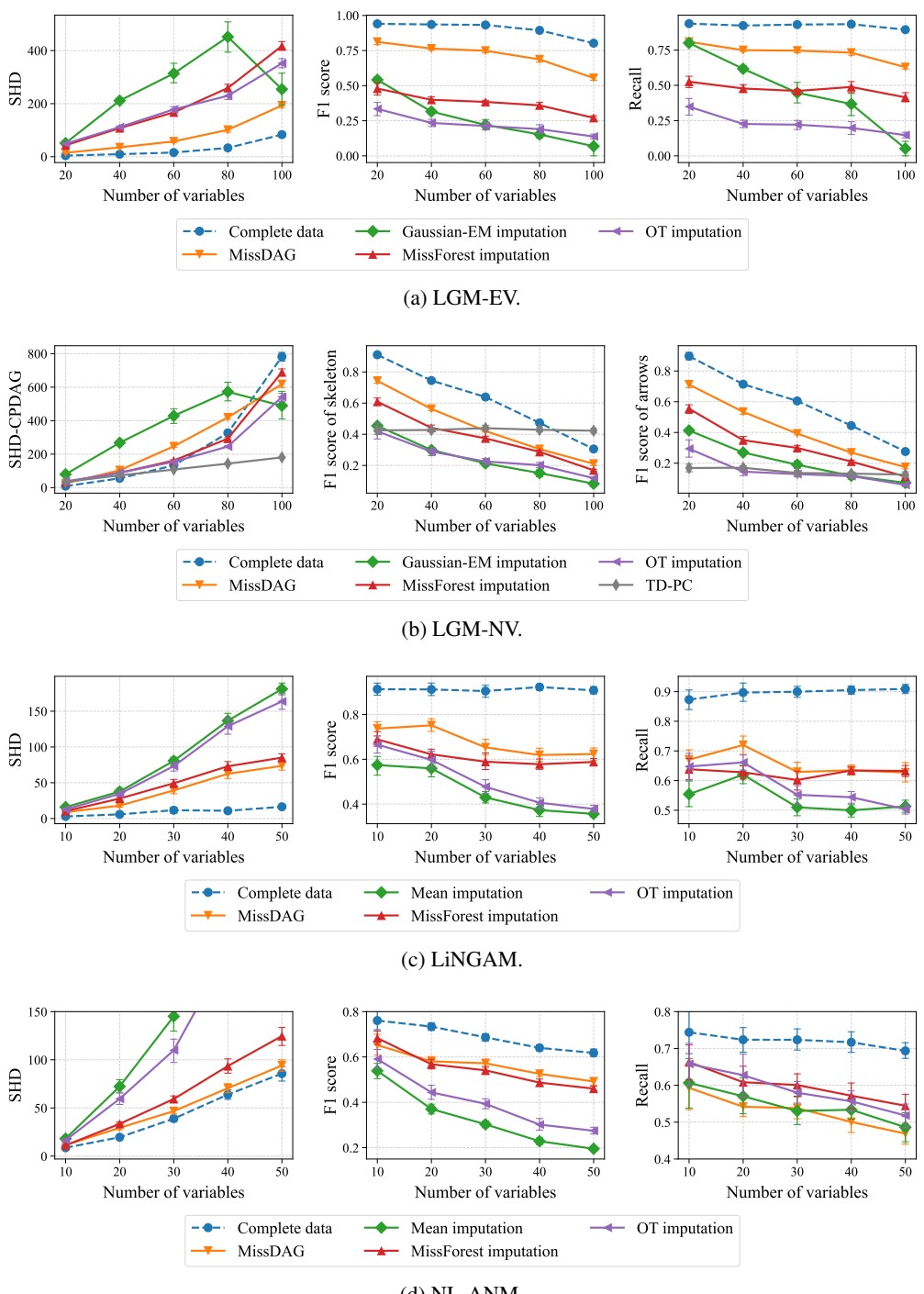

(a) LGM-EV.

(b) LGM-NV.

(c) LiNGAM.

(d) NL-ANM.

Figure 9: Results on different numbder of nodes.

## F.5 Scalability of different nodes

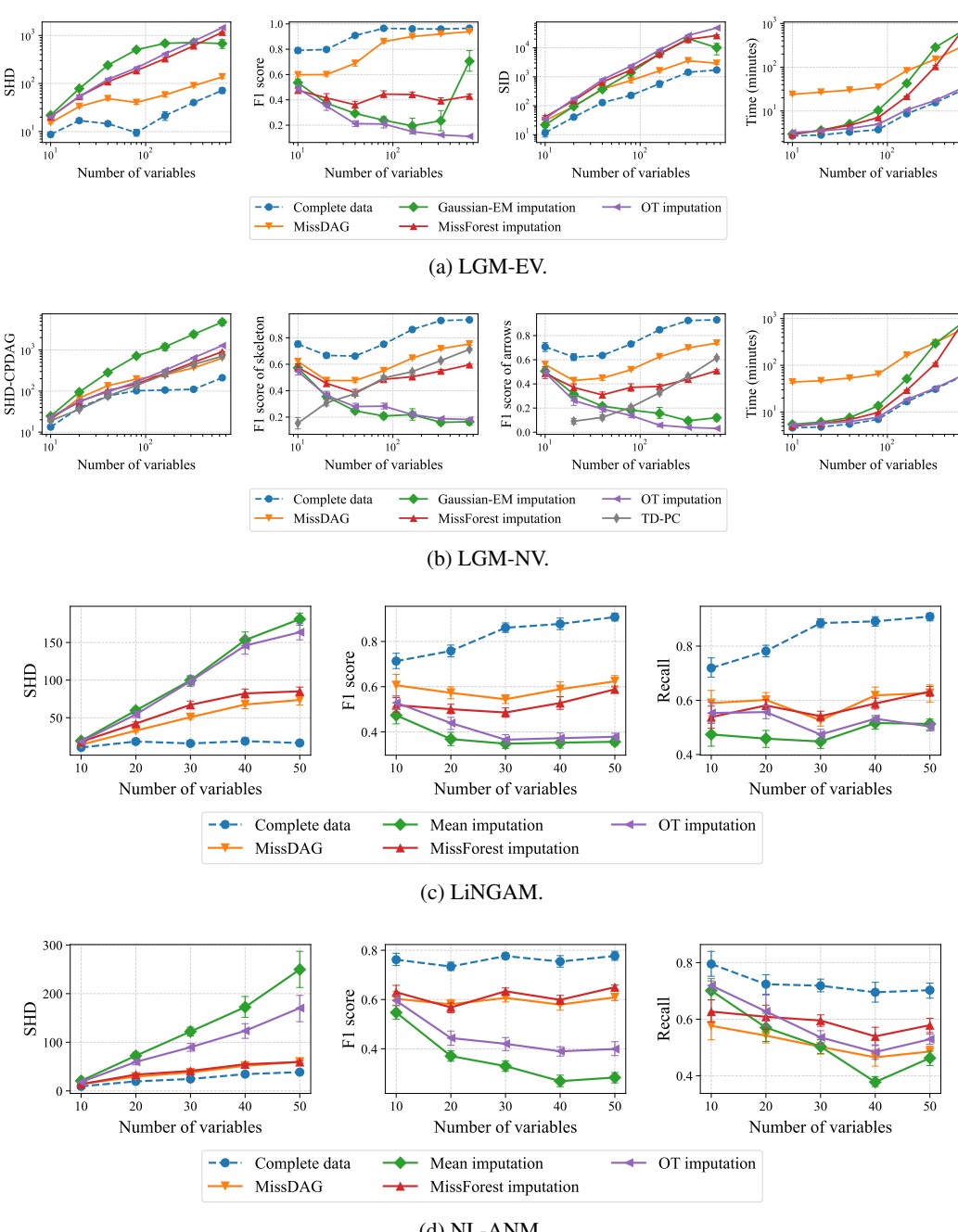

(a) LGM-EV.

(b) LGM-NV.

(c) LiNGAM.

(d) NL-ANM.

Figure 10: Results on the scalability of different nodes ($2d$ samples for $d$ nodes).

## F.6 Different degrees

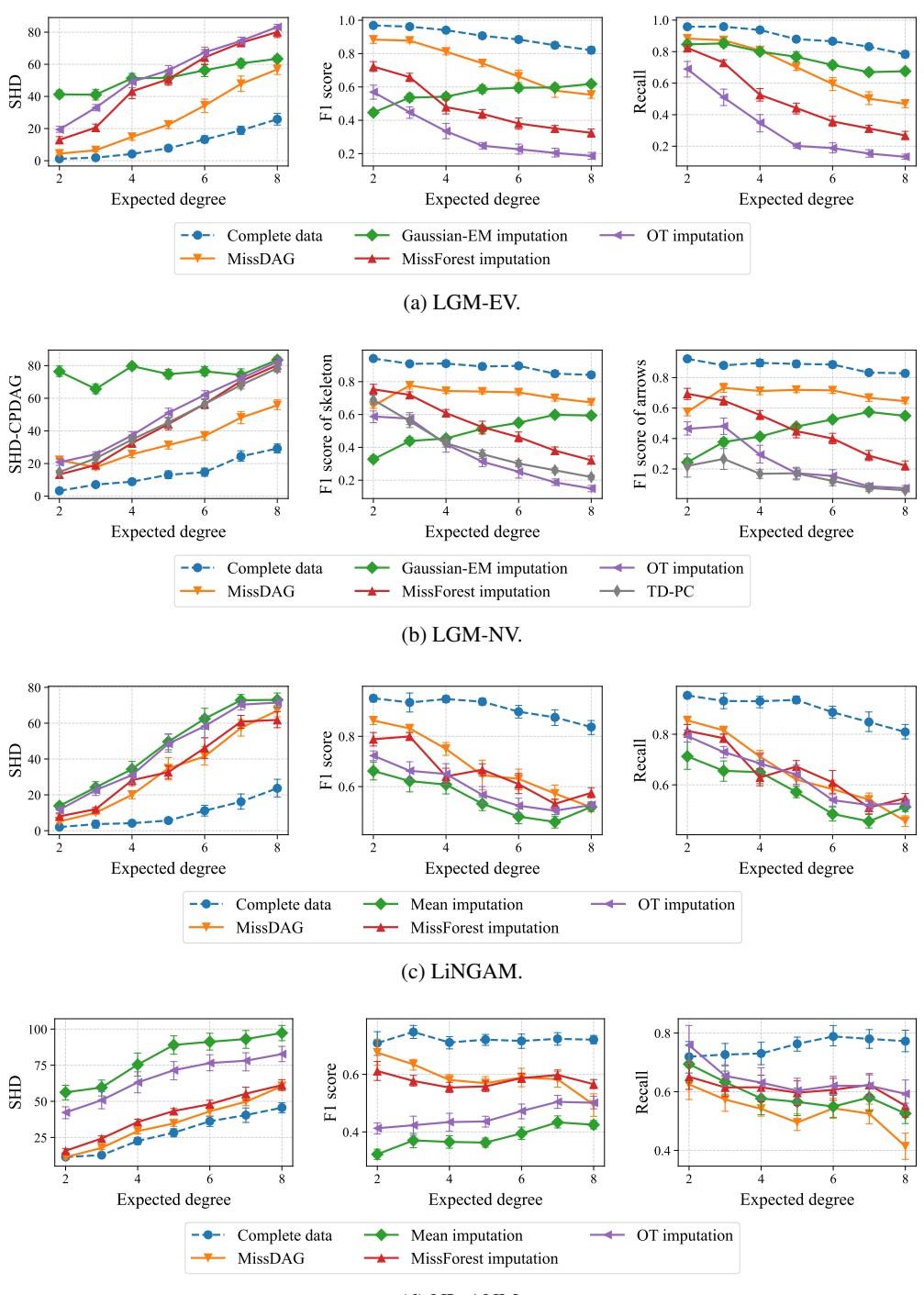

(a) LGM-EV.

(b) LGM-NV.

(c) LiNGAM.

(d) NL-ANM.

Figure 11: Results on different degrees.

## F.7 Related to LiNGAM

### F.7.1 Different noise types

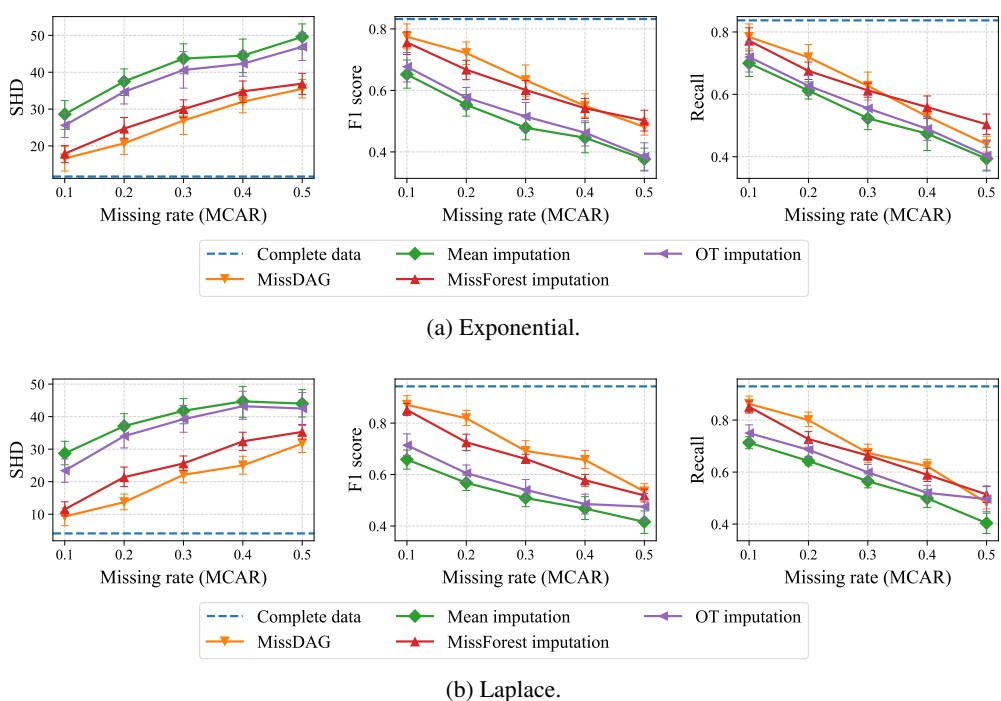

(a) Exponential.

(b) Laplace.

Figure 12: Results of LiNGAM on different noise types.

### F.7.2 Different causal discovery methods (ICA-LiNGAM & Direct-LiNGAM)

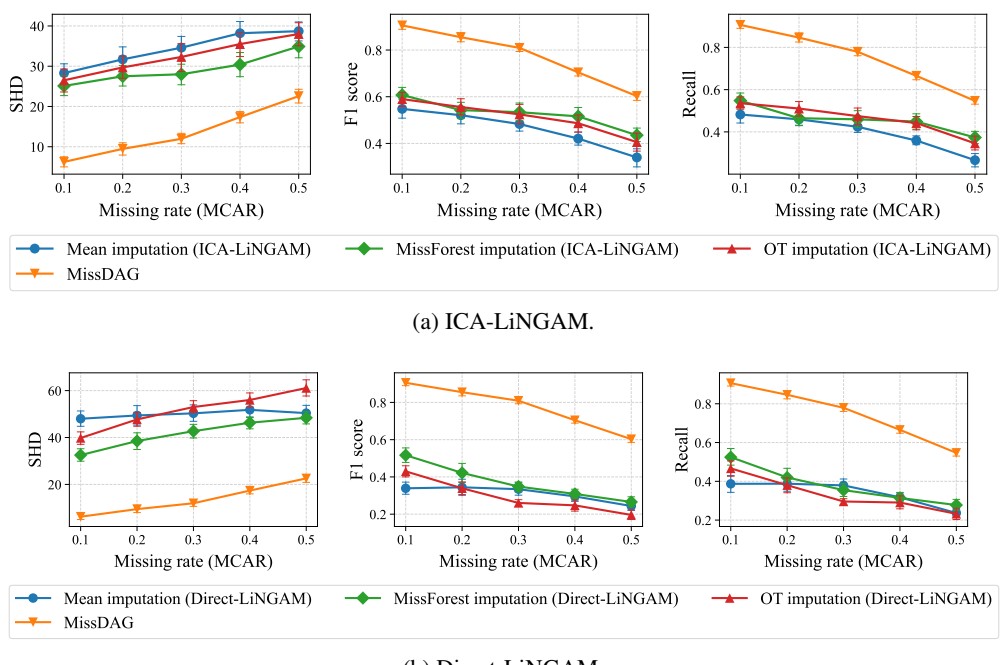

(a) ICA-LiNGAM.

(b) Direct-LiNGAM.

Figure 13: Results of LiNGAM with different causal discovery baseline methods.

## F.8 Different non-linear types (NL-ANM)

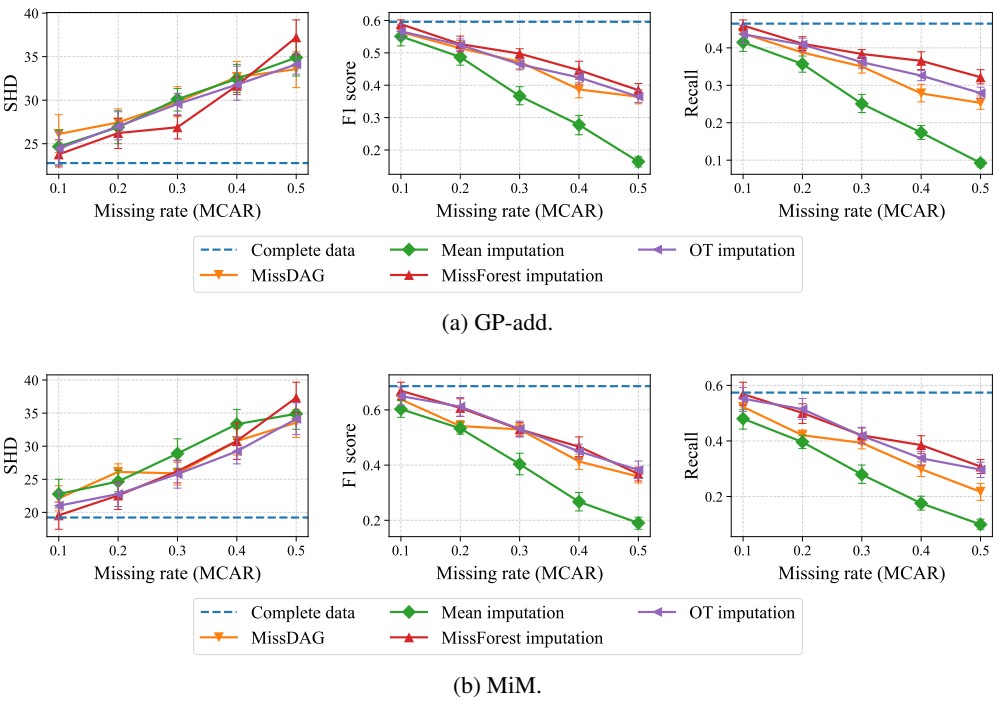

(a) GP-add.

(b) MiM.

Figure 14: Results of NL-ANM on different non-linear functions.

# G Running time

The running time of the proposed methods and the baselines are shown in Figure 10. We observe that the proposed method has a longer running time when the number of variables is small, which may not be surprising because, different from the other one-stage imputation methods, our proposed method inherits the iterative optimization property of EM method, leading to a longer running time. Nevertheless, as soon as the number of variables gets very large (i.e., more than 640 variables), our method runs faster than the other strong baselines (i.e., Gaussian-EM and MissForest imputations) since these methods also take much time for imputation. Moreover, the running time of the other baselines appears to increase more quickly w.r.t. the number of variables as compared to our proposed method. These observations indicate that our method appear to scale well, in addition to the improvement of structure learning performance observed in the experiments.