# OpenReview forum: "MissDAG: Causal Discovery in the Presence of Missing Data with Continuous Additive Noise Models"
_NeurIPS.cc/2022/Conference — NeurIPS 2022 Accept_

### Official Review · Reviewer_vkvi · 2022-07-06

**Rating:** 6
**Confidence:** 4
**Soundness:** 3 good
**Presentation:** 2 fair
**Contribution:** 2 fair

**Summary:**

This paper studies the problem of causal discovery when parts of the observation matrix are missing. The proposed method called “MissDAG” is an EM-like procedure: the E-step estimates the expected data log-likelihood as a function of the unknown causal graph and parameters by marginalizing out/sampling the missing values given the graph and parameters estimated in the previous step; the M-step then maximizes this expected data log-likelihood function with respect to the unknown causal graph and parameters. In an empirical evaluation, the work shows that MissDAG results in better graph recovery in terms of SHD/F1 scores than using other imputation methods.

**Questions:**

Method:

A main question arising in Section 4 concerns the likelihood term $p(\mathbf{X} | \theta)$ (Section 4.1.1). Throughout the paper, we assume that the SCMs considered are *acyclic* (line 74 and 79). Thus, irrespective of the functional form and the noise distribution, the likelihood factorizes into a product of local conditionals $p_j(x_j | pa_j)$ (the Markov kernels). As a result, the likelihood is always trivial to evaluate conditional on the causal graph and parameters, especially for the additive noise models considered in this work.

Hence, I do not understand why this work introduces the complicated generalization in Eqs. (7)-(8) involving determinants and the Jacobian. This general form was used by [32] to handle *cyclic* SCMs, in which the likelihood is not trivially computable in the above, recursive form. However, the cyclic setting is not studied in this work.

Later in the manuscript (Proposition 1, line 216), the authors even note that for a DAG, this additional term is 1 and can thus be dropped from Eqs. (7)-(8), which reduces the likelihood computation to the simple factorized computation always used in BN structure learning (This is even explicitly spelled out in lines 217-218). Unless I missed something essential about MissDAG and the incomplete data setting, this unnecessary complication in the likelihood somewhat indicates of a lack of familiarity with related work and the task of causal structure learning more generally. The whole of Section 4 could be significantly simplified and thus made more accessible when this is dropped.

Related to the above, I do not see why Eq. (10) needs to be so complicated. The likelihood in the linear Gaussian is trivial to evaluate and simply a product of Gaussian distributions with densities that are tractable to compute, without needing to introduce $\mathbf{T}$. It would also be straightforward to add a bias term to the linear Gaussian model instead of using the structural equation model notation in Eq (9), where each causal variable has zero mean.

Related work:

Please clarify the precise relationship to and contribution over the prior works [15] and [54]. It appears that their algorithms conceptually do the same EM procedure as MissDAG, except that MissDAG substitutes in more modern likelihood model assumptions and structure learning methods into the subcomponents of the algorithm. For instance, [54] states: “[Our algorithm] uses the current estimate of the structure and the incomplete data to refine the conditional probabilities, then imputes new values for missing data points by sampling from the new estimate of the conditional probabilities, and then refines the network structure from the new estimate of the data using standard algorithms for learning Bayesian networks from complete data.”


**Limitations:**

Overall, the paper presents an intuitive idea based on EM for the causal discovery problem from incomplete data, which is validated convincingly in experiments on simulated data but is not new in the literature [15, 54]. In addition, given the points brought up above, the presentation and derivation have room for improvement both in terms of writing and in terms of clarity and motivation of the mathematical derivations, which sometimes indicates a lack of familiarity with existing approaches to the problem (in particular, the point on computing the likelihood). This makes an otherwise clear and intuitive method, which appears to be effective empirically, very hard to parse and internalize. Since the idea and results are convincing overall, I am open to revising my scores if the issues raised have been addressed.

**Strengths And Weaknesses:**

The motivation of the paper and the introduction are convincing. It makes sense to use EM to account for missing values in this context, and it has been done by previous work (e.g. [15, 54]). The array of imputation baselines and considered settings are likewise comprehensive and show the efficacy of MissDAG in the context of causal discovery. Since we are interested in the _causal_ graph, it would make sense to also show metrics like the structural intervention distance (SID), which is closer to our interests in causal inference.

While the overall presentation structure of the paper is fine, the presentation of the various subparts and subsettings in the methods section (Section 4) is quite convoluted, making it hard to parse the key challenges and their proposed solutions. I think the exposition could be improved by first introducing the E-step and M-step and their different levels of difficulty (exact vs. approximate posteriors) without reference to a specific generative model yet (such as LGM, LiNGAM, NL-ANM). Then, to separate the MissDAG algorithm from the specific generative model assumptions, specific instantiations of these steps could be presented in an additional Section 5 that gives certain closed forms, as for instance in the linear Gaussian case.

Further points/questions about the presentation:
- Figure 3: “$\Sigma$ (sufficient statistic) can be a criterion for distribution recovery.” What is your justification for that? In this paper, the goal is causal structure learning and not (observational) distribution recovery, so the work should give some motivation to show this figure and metric.
- Line 46: “the consistent approach”. What is the consistent approach that is meant here? This is related to the previous point about Figure 1, which is referenced in this sentence.
- Figure 6(b): What is a “history”? This is never explained anywhere.
- In line 56, you mention the concept of “ignorability” when stating your main contribution, even though this concept was never introduced at this point of the manuscript. It would be good if a sentence earlier in the introduction explains this notion and why it is needed.
- I do not understand the point of introducing Eq. (2) and/or $\phi$, since it is never used in the later sections of the paper. Can you provide some motivation? Also, why is “ignorable missingness” (Assumption 1, line 95) important? This is not motivated explicitly and never picked up again later in the manuscript, and unless the reader is very familiar with this literature already, it may be difficult to relate to the other parts of the paper.
- Line 124: What is “$\theta_\mathcal{M}$”? This notation is not used anywhere else.
- Line 128: What do you mean by imputing “the missing entries by the functions of $X_M$”? Do you mean the distributions of $X_M$?
- Line 168-169: “we also plug in the DAG constraint as an inductive bias for optimization.” The DAG constraint is hardly an inductive bias, more a hard constraint (line 74), which usually makes the structure learning problem harder to solve and not easier. Please consider rephrasing.
- Line 189-190: “… the exact formulation of each noise distribution is agnostic.” What is meant by this? This is most likely a wrong usage of the word “agnostic”.
- Line 205-206: This is imprecise. You should indicate that $x^j_{im}$ are samples from the posterior $p(\mathbf{X}_{m} | X_o, \theta^t)$.
- Lines 299-300: “Different from the BN learning methods, our method is specifically designed for causal discovery, where identifiability of the underlying causal graph is critical.”
I do not agree with this statement. There is no inherent difference between causal discovery algorithms from observational data using a score- or constraint-based approach and algorithms for learning a BN. For certain classes of BNs (such as the ANM ones studied in this work), the true causal structure achieves highest likelihood, which is why these models are deemed “identifiable” from observational data. In fact, NOTEARS itself is a method for BN structure learning and used pervasively in this work as a causal discovery algorithm.

Typos and minor suggestions:
- Abstract (lines 6-7): “… a two-step method may suffer from suboptimality, as the imputation algorithm is unaware of the causal discovery step.”
This sentence is vague, especially since MissDAG is not “aware” of the causal discovery step either: MissDAG uses an estimated causal model to approximate the expected data log-likelihood. What does it mean to be “aware” of the causal discovery step? Formulations about efficiency and/or bias as in lines 42-45 would be more convincing.
- Line 21: “concerned variables” -> consider rephrasing as, e.g. “variables of interest”
- Line 83: typo, the second $\mathbf{X}$ should be a $\mathbf{Y}$
- Line 87: typo, I believe “observed portions” should be “observed positions”
- Line 118: “a graph learning part $\theta_\mathcal{G}$”. It appears as though this is the causal graph. If so, it may be clearer to use $\mathcal{G}$ as the notation (similar to line 75 or 171), since the causal graph is usually considered to be separate from the parameters or functions parameterizing an SCM.
- Line 119: “While in …, … .“ This sentence is grammatically incorrect.
- Line 125: “which relates the estimation … to the same parameters estimation”. This sentence is grammatically incorrect.
- In my opinion, the related work section would work better as Section 3. Otherwise, it is not transparent from the beginning what the gap in the state of the art is that MissDAG sets out to fill, especially since EM-like approaches have previously been used for BN structure learning.

---

> ### Author Response · Authors · 2022-08-02
> **Responses to Reviewer vkvi [Q5-6]**
>
> **[Q5. Acyclic likelihood, log-determinant, and Jacobian in Eqs. (7), (8), (10).]** Thanks for this very insightful comment. We totally agree with your comments, summarized as:
> - The log-determinant term can be dropped if the search space can be guaranteed to be DAGs, because, as you also noticed, the log-determinant term becomes zero if the candidate solution is acyclic (Proposition 1).
> - Eqs. (7), (8), (10) can be simplified if we drop this log-determinant term, which reduces to the simple factorized likelihood (that corresponds to likelihood based on DAGs) always used in BN structure learning.
>
> The first point is especially relevant if *discrete* score-based structure learning methods (e.g., A*, dynamic programming) are applied since the candidate solutions are acyclic. At the same time, we would also like to point out that some recent score-based structure learning methods based on *continuous optimization* have shown that including the log-determinant term (which corresponds to likelihood based on directed cyclic graphs) in its objective function may be desirable and lead to better performance (although there also exist continuous methods that do not include the log-determinant term such as NOTEARS). For instance, a continuous structure learning method GOLEM [38] was recently proposed which utilizes the likelihood based on cyclic graphs and thus includes log-determinant; furthermore, NOTEARS-ICA [74] also includes the log-determinant term in its objective function. Therefore, we aim to develop a general framework that flexibly includes both classes of these existing discrete and continuous likelihood-based methods, with or without the log-determinant term; see Figure 2.
>
> Therefore, if possible, we would still prefer to use the current form of the likelihood that includes the log-determinant term. Given your comment, we have included a summary of the above discussion in Section 4.1.1 of the revision and also moved Proposition 1 to Section 4.1.1 to improve the clarity. Hope this addresses your concern and please kindly let us know if you have further concerns.
>
> **[Q6. BN learning and causal discovery in Line 299-300 \& contributions over the prior works.]** Thanks for the thoughtful comment. We agree with your opinion on the similarities between BN structure learning and causal discovery from observational data. To address your concern, we have rephrased this sentence to *Existing BN learning methods from incomplete data focus on identifying the Markov equivalence classes (i.e., discrete cases) under suitable assumptions and usually formulate the structure learning problem as a discrete optimization program, while our work focuses on continuous identifiable ANMs of which the structure is fully identifiable and includes recent structure learning approaches based on continuous optimization.*
>
> Furthermore, as you suggested, our work indeed shares similarities with [15, 58], since both rely on the EM algorithm. However, as described above, our work focuses on continuous identifiable ANMs that have recently received considerable attention, while [15, 58] focuses on discrete cases in which one is only able to identify the Markov equivalence class; therefore, the key technical development is different. (1) For the linear Gaussian case, we derive the closed-form solution of exact posterior that is different from the discrete case considered by [15, 58]. (2) For the linear non-Gaussian and nonlinear cases, since the exact posterior is not available in closed form, we develop a method based on approximate posterior using Monte Carlo and rejection sampling; such a setup that involves approximate posterior may be more challenging and has not been considered by [15, 58], since the exact posterior of the discrete case considered by [15, 58] is available in closed form. (3) As pointed out by the reviewer and described in our response to Q5, our formulation may be considered more "general" (e.g., the log-determinant term is incorporated in the likelihood function) and include modern structure learning approaches based on continuous optimization (in addition to classical methods based on discrete optimization considered by [15, 58]). We hope this addresses your concern and have added this discussion in the supplementary related work section (see Appendix A) to make our contribution clearer.

---

> > ### Comment · Reviewer_vkvi · 2022-08-05
> > **Score update**
> >
> > Thank you for your very detailed response.
> >
> > First, thanks for the detailed explanation regarding the log determinant term. This makes sense to me, and the restructuring of 4.1.1 and the additional paragraph helped me understand this part better. However, since your method clearly intends to target acyclic graphs/SCMs (both in terms of the title, exposition, and experiments), I am still not sure I see the motivation of the term in the first place. The point of performance only gives a partial motivation since you do not provide any evidence that the log determinant term helps empirically. If the goal of the paper is also to solve the cyclic case, then this motivation has to be made much clearer throughout the paper. If this were the case, more care must also be taken to make SCMs well-defined in Section 2.
> >
> > The updated related work is better and makes clearer that the distinguishing aspect to prior work is the continuous optimization and the model class and not the EM approach. Re. Figure 1: Thanks, I now see your point.
> >
> > Overall, various aspects of the work were improved. Thus, I am increasing my score to 6.

---

> > > ### Author Response · Authors · 2022-08-06
> > > **Remove the log-determinant term.**
> > >
> > > Thanks for the kind suggestion.
> > >
> > > We totally agree with your opinion that it will be better to remove the log-determinant term because of the unclear motivation. In our paper, we still restrict our attention to the acyclic case and leave the cyclic setup in future work. In the revision, we have deleted this term and clarified our acyclic setup. Thanks again for your time and great efforts.
> > >
> > > Authors

---

> ### Author Response · Authors · 2022-08-02
> **Responses to Reviewer vkvi [Q3-4]**
>
> **[Q3. Questions about the presentations.]**
> - **Figure 1:** Thanks for asking this insightful question, which is exactly what we are trying to highlight. In Figure 1, we compare the distribution recovery performances by $\Sigma$ and structure learning abilities of Gaussian-EM imputation and MissDAG. Our goal is to demonstrate the key difference between the structure learning task and the distribution recovery task. Even though methods like Gaussian imputation is able to recover the underlying data distribution very well (since there is no model misspecification), its structure learning performance appears to be suboptimal, because, as you nicely pointed out, distribution recovery is different from structure learning. The reason is that the two-stage Gaussian-EM method first uses the EM algorithm to impute the data assuming Gaussian distribution, and then performs DAG estimation. In this case, the information of acyclicity and equal noise variances is not utilized in the first stage (imputation); therefore, the recovered empirical covariance matrix, despite being closer to the true one, may correspond to an entirely different DAG, thereby degrading the performance of DAG estimation in the second stage. This is also an important motivation for our work, i.e., developing the MissDAG framework that alleviates this issue by combining the two stages. We have clarified this in the revised paper.
>
> - **"The consistent approach" in Line 46:** By "the consistent approach", we intend to convey that there is no model misspecification between the true (Gaussian) distribution and the Gaussian-EM imputation method; therefore, Gaussian-EM imputation is able to recover the data distribution given enough data, which thus is "consistent". We have rephrased this sentence to *Moreover, as shown in Fig. (1), even though the Gaussian-EM imputation method can consistently recover the data distribution as there is no model misspecification, it may lead to a sub-optimal directed acyclic graph (DAG) estimation because it focuses solely on distribution recovery instead of structure learning.*
>
> - **"History" in Figure 6(b):** $W^{(n)}$ refers to the estimated graph $\theta_{\mathcal{G}}$ in the $n$-th EM iteration. We have clarified this in the revision.
>
> - **Concept of "ignorability" in Line 56:** Following your suggestion, we have changed the statement to "in which the underlying missing mechanism is independent from the observed information" and defer the explanation of "ignorability" to Section 3.
>
> - **Motivation of Eq. (2) and "ignorable missingness":** Ignorable missingness (Little and Rubin, 2002) is important as it is a common assumption that is required by EM-style algorithms and also our method. It can be interpreted as a belief that the available data is sufficient to "correct" the missing data, by assuming that the missingness and model parameters are distinct. Furthermore, the full likelihood in Eq. (2) (with parameter $\phi$) is needed as a context to motivate/derive the ignorable likelihood in Eq. (3), which is the backbone of our method in Section 4. We have made these explicit in the revision.
>
> - **$\theta_\mathcal{M}$ in Line 124:** Following your comment, we have changed $\theta_\mathcal{M}$ to "the parameters of the SCM" in the revised paper.
>
> - **Imputing the missing entries by the functions of $X_M$:** We have changed "functions of $X_M$" to "the distribution of $X_M$" according to your suggestion.
>
> - **DAG constraint as inductive bias in Line 168-169:** Thanks for pointing this out which helps improve the clarity. Following your suggestion, we have rephrased the sentence to *Here, we also plug in the DAG constraint for optimization to ensure that the estimated graph is acyclic.*
>
> - **Usage of "agnostic" in Line 189-190:** We have changed "agnostic" to "unknown".
>
> - **Line 205-206 is imprecise:** According to your suggestion, we have changed to "$x^j_{im}$ are samples from the posterior".
>
> **[Q4. Typos and minor suggestions.]** Following your suggestion, we have fixed these typos and moved some related work before our method part in the revision.

---

> ### Author Response · Authors · 2022-08-02
> **Responses to Reviewer vkvi [Q1-2]**
>
> We greatly appreciate the reviewer's thorough and constructive comments, many of which will help improve the clarity and presentation of our paper. We attempt to address all the concerns in the following.
>
> **[Q1. SID metric.]** Thank you for this helpful suggestion. Following your suggestion, we have included the structural intervention distance (SID) for the linear Gaussian case in the scalability experiment; see the third panel of Figure 11a in Appendix F.5 of the revision. We found that this metric yields consistent observations with the existing metrics reported in the paper. Specifically, our method has lower SID (i.e., better performance) as compared to the baselines, especially when the number of variables increases. We will report this metric for other settings as well in the final version of the paper.
>
> **[Q2. Re-organization.]** Thanks for your constructive suggestion, which would help to make our method clear. We are trying our best to clarify our current layout by referring to some of your kind suggestions (still working) and will keep improving the manuscript to make it clear and logical.

---

### Official Review · Reviewer_sgce · 2022-07-06

**Rating:** 7
**Confidence:** 4
**Soundness:** 3 good
**Presentation:** 3 good
**Contribution:** 3 good

**Summary:**

The method proposed copes with the missing data problem in causal discovery by maximising the expected likelihood of the visible part of observations using the expectation-maximisation framework.

**Questions:**

Questions. In Section 3, in notations,  X \in R^{N \times d} is repeated twice; the second should be Y?

I found it not easy to follow Section 4.1.1. It is stated that "benefiting from the well-researched results of density transformation...": it might be well-researched but not extremely well known, and it is not easy to understand where equation 7 comes from.

I am not sure that the notation in equation 8, where X comes twice (on the left and on the right) is successful.

The explanation what "SHD" stands for comes a little bit late in the text.


**Limitations:**

I guess that the method is not scalable (since it relies on the EM).
From the causal viewpoint: it is assumed that there is not any latent confounders and there are no cycles in the graph.

**Strengths And Weaknesses:**

Strengths. The paper is well written. The technical details are provided, and the results of numerical experiments are convincing.

Weaknesses. The EM algorithm to treat missing data is quite a standard approach. Its extension to the current problem is rather straightforward. Difficult cases, such as latent common causes or cyclic graph are not considered.

A DAG is not necessarily a causal graph, as far as I know. Section 4 is focused on an extension of EM to DAGs. It seems that there is nothing specific for causal inference in this contribution.

In Related Work: the papers cited are quite old. There is much recent work on missing data processing (but not necessarily for DAGs).

---

> ### Author Response · Authors · 2022-08-02
> **Responses to Reviewer sgce [Q6-8]**
>
> **[Q6. $X$ comes twice in Eq. (8).]** We have carefully checked Eq. (8) and believe that it is the right form. Intuitively speaking, the term $X_{ij}-f_i(X_i)$ can be loosely interpreted as "neighborhood regression", which regresses each variable on the other variables (since the acyclicity constraint will discard self-loops). If there is any misunderstanding, please kindly let us know.
>
> **[Q7. SHD.]** We introduced the abbreviation "SHD" in Section 5. Given your comment, we have provided the definition of SHD of the revision, which refers to the smallest number of edge additions, deletions, and reversals required to transform the recovered DAG into the true one. Thanks.
>
> **[Q8. Scalability.]** As the reviewer nicely pointed out, our method inherits the time inefficiency issue of the EM algorithm. The experiments in Appendix F.5 demonstrate that our method appears to scale up to $640$ variables in the linear Gaussian case, and up to $50$ variables in the linear non-Gaussian and nonlinear ANM cases. The reasons are that (1) samplings methods are adopted to compute the approximate posterior in the latter two cases, which lead to a longer running time, and (2) when the exact posterior is available in the linear Gaussian case, sampling methods are not needed and thus the overall method runs much faster and scales well. Therefore, a future direction is to explore variational inference to approximate the posterior to scale up to larger problems by sacrificing some accuracy. We have included a discussion of this limitation and future work in Section 6 of the revision.
>
> **Reference**
>
> A. Ghassami, A. Yang, N. Kiyavash, and K. Zhang. Characterizing distribution equivalence and structure learning for cyclic and acyclic directed graphs. In International Conference on Machine Learning, 2020.
>
> R. Bhattacharya, T. Nagarajan, D. Malinsky, and I. Shpitser. Differentiable causal discovery under unmeasured confounding. In International Conference on Artificial Intelligence and Statistics, 2021.
>
> Imke Mayer, Julie Josse, Félix Raimundo, and Jean-Philippe Vert. Missdeepcausal: Causal inference from incomplete data using deep latent variable models. arXiv:2002.10837, 2020.
>
> Chao Ma and Cheng Zhang. Identifiable generative models for missing not at random data396 imputation. Advances in Neural Information Processing Systems, 2021.
>
> Tomas Geffner, Javier Antoran, Adam Foster, Wenbo Gong, Chao Ma, Emre Kiciman, Amit Sharma, Angus Lamb, Martin Kukla, Nick Pawlowski, et al. Deep end-to-end causal inference. arXiv:2202.02195, 2022

---

> ### Author Response · Authors · 2022-08-02
> **Responses to Reviewer sgce [Q1-5]**
>
> We are grateful for the reviewer's effort and the positive comment on our paper. Please find the response to your questions below.
>
> **[Q1. Latent confounder and cyclic.]** Thanks for pointing this out. Causal discovery is a challenging ill-posed problem, and some assumptions, e.g., causal sufficiency, is commonly adopted for the initial development of a practical approach, e.g., PC, GES, LiNGAM. It remains an active research area to relax this assumption which has attracted much attention. To deal with latent confounders, we plan to extend our framework to handle hidden confounders in future work, e.g., by incorporating the method by Bhattacharya et al. (2021) designed for ancestral ADMGs. Similarly, extending the proposed method to learn cyclic graphs from incomplete data will also be interesting for future work. We believe that the likelihood-based (cyclic) structure learning method developed by Ghassami et al. (2020) can be applied. We have discussed it in Section 6 of the revision.
>
> **[Q2. Nothing specific for causal inference.]** Great point. Indeed, there may be a difference between a DAG and a causal graph--the directed edges of the latter are given a causal meaning that allows it to answer interventional queries. Causal discovery and structure learning may be interpreted as a prior step to learning the structure before causal inference tasks (e.g., treatment effect estimation) can be carried out (if prior knowledge of such structure is not available), although in many cases causal discovery methods from purely observational data do not explicitly make use of such ability to reason about interventions. We have included a discussion about it in Section 2.
>
> **[Q3. Recent works.]** Thanks for the suggestion. We will update the related work section in the final version to discuss more recent works related to missing data processing, such as MissDeepCausal (Mayer et al. arXiv2020), GINA (Ma et al. NeurIPS21), DECI (Geffner et al. arXiv2022) and more progress in causality with incomplete data. If there is any specific reference the reviewer has in mind that we are encouraged to discuss, please kindly let us know.
>
> **[Q4. $X^{N\times d}$ typo.]** Thanks for spotting this. We have fixed this typo in the revision.
>
> **[Q5. Reference of Eq. (7).]** Thanks for asking this question which helps improve the clarity of the paper. Eq. (7) comes from the change of variables rule of density transformation (i.e., from noises $Z$ to variables $X$), as shown in [35]. We have made this clear in the revision to make it easier for readers to follow.

---

### Official Review · Reviewer_woaA · 2022-07-11

**Rating:** 5
**Confidence:** 4
**Soundness:** 3 good
**Presentation:** 3 good
**Contribution:** 2 fair

**Summary:**

The paper proposes a method called MissDAG for learning causal structures in the presence of missing data. It assumes the data is missing at random (MAR) and the underlying causal model is an additive noise model (ANM). The paper derives expectation-maximization (EM)-based algorithms for maximizing a penalized log-likelihood score under different classes of ANMs including the linear non-Gaussian models, linear Gaussian models with equal noise variances, and nonlinear ANMs. The proposed MissDAG method is evaluated through synthetic and real data experiments.


**Questions:**

Why are MissDAG methods not empirically compared with existing work on causal discovery or Bayesian network (BN) structure learning from missing data?

What are the sizes of the models (the number of variables $d$) in LiNGAM and NL-ANM experiments shown in Fig. (3)?

What is the running time of MissDAG vs. baselines in the experiments?

How well does MissDAG scale to larger models?


**Limitations:**

The paper did not discuss the limitations of the proposed approach. One limitation I see is that the proposed approach may be too slow to learn large models which I believe is a well-known issue in the EM-type of algorithms for learning BN structures from missing data. The largest models tested in the experiments contain 20 variables. How well does MissDAG scale to larger models?


**Strengths And Weaknesses:**

The paper addresses an important problem of causal discovery in the presence of missing data. The existing work mostly focuses on structure discovery from discrete data while this work studies learning causal structures from incomplete data under ANMs. The proposed technical approaches are mostly standard techniques – its main contributions are deriving EM steps under different classes of ANMs.

The experiments only compared with imputation methods as baselines, but not with existing work on causal discovery or Bayesian network (BN) structure learning from missing data, e.g. Structural EM [16], MVPC [60].  Why are MissDAG methods not empirically compared with existing works?

The writing is overall clear, but there are grammar issues here or there.

---

> ### Author Response · Authors · 2022-08-02
> **Responses to Reviewer woaA**
>
> We sincerely thank the reviewer for the helpful feedback and time devoted. Below we give a point-by-point response to the comments.
>
> **[Q1. Comparisons with Structural EM and MVPC.]** Thank you for asking this question. Both Structural EM [15, 16] and MissDAG adopt the EM-based framework for learning Bayesian networks from incomplete data. However, in [15, 16], only methods for discrete data are developed, whereas we focus on identifiable continuous ANMs in our paper. Therefore,  structural EM and our method are not directly comparable.
>
> MVPC [64] focuses on finding the identifiable conditions and extending the PC method for the Missing Not At Random (MNAR) case, whereas our work focuses on the *finite sample* problem for the Missing At Random (MAR) case, which is not considered by MVPC. Nonetheless, following your suggestion, we have conducted further experiments and included the empirical results of MVPC in Figure 8b of Appendix F.2 in the revision, which shows that our method performs better than MVPC in all cases. It is worth noting that even though there is no theoretical guarantee of EM algorithm for MNAR, empirical results show that our method performs better than the other baselines including MVPC. We will include this discussion in the final version of the paper.
>
> **[Q2. Size of models/number of variables in Fig. (3).]** The number of variables is $20$ for LiNGAM and NL-ANM. We have clarified this in the revised paper.
>
> **[Q3. Running time.]** The running time of the proposed methods and the baselines are shown in Figures 11a and 11b in Appendix F.5. We observe that the proposed method has a longer running time when the number of variables is small, which may not be surprising because, different from the other one-stage imputation methods, our proposed method inherits the iterative optimization property of EM method, leading to a longer running time. Nevertheless, as soon as the number of variables gets very large (i.e., more than $640$ variables), our method runs faster than the other strong baselines (i.e., Gaussian-EM and MissForest imputations) since these methods also take much time for imputation. Moreover, the running time of the other baselines appears to increase more quickly w.r.t. the number of variables as compared to our proposed method. These observations indicate that our method appears to scale well, in addition to the improvement of structure learning performance observed in the experiments in Appendix F. We have included this discussion in Appendix G.2 of the revision.
>
> **[Q4. Larger models \& limitations.]**
> As the reviewer nicely pointed out, our method inherits the time inefficiency issue of EM algorithm. The experiments in Appendix F.5 demonstrate that our method appears to scale up to $640$ variables in the linear Gaussian case, and up to $50$ variables in the linear non-Gaussian and nonlinear ANM cases. The reasons are that (1) sampling methods are adopted to compute the approximate posterior in the latter two cases, which lead to a longer running time, and (2) when the exact posterior is available in the linear Gaussian case, sampling methods are not needed and thus the overall method runs much faster and scales well. Therefore, a future direction is to explore variational inference to approximate the posterior to scale up to larger problems by sacrificing some accuracy. We have included a discussion of this limitation and future work in Section 6 of the revision.
>
> **[Q5. Grammar issues.]** We will carefully proofread our paper to fix the grammar issues in the revised paper.

---

### Official Review · Reviewer_GKqy · 2022-07-12

**Rating:** 6
**Confidence:** 3
**Soundness:** 3 good
**Presentation:** 2 fair
**Contribution:** 3 good

**Summary:**

An EM-based method is described for doing causal discovery when some data entries are missing (MAR or MCAR). Different versions are proposed, which can deal with different variants of additive noise models. Experimental results show it tends to improve over imputation methods.

**Questions:**

* What can you say about the running time of your method compared to other methods for the same task?

* The conclusion of appendix E (Convergence analysis) was unclear to me. Under what conditions is convergence guaranteed? Does this suffice to guarantee overall correct behaviour of the method (e.g. in the sense of finding the same graph as would be found without missingness, given enough data)?

Minor points:
* line 77: "include" -> "includes"; I recommend adding a comma before "and"
* line 78: "is no" -> "are no"
* line 80: "indicate" -> "indicates"
* line 83: 2nd formula about X should be about Y, and in {0,1}$^{N \times d}$
* line 106: "applys" -> "applies"
* line 115: "follow the fashion that": pick another word in place of "fashion" (then make the sentence grammatical)
* line 124: $\theta_{\mathcal{M}}$: is this a new notation?
* line 178: NL-ANM and LiNGAM are used without having been introduced, please mention the abbreviation when the work is cited earlier.
* line 231: "Completely Partial DAG" -> "Complete Partial DAG"
* line 278: I guess the "is" should be removed

**Limitations:**

-

**Strengths And Weaknesses:**

The problem is practically relevant and the theoretical analysis looks sound, though I did not check it closely. I think the clarity of the paper suffers somewhat from the large amount of material compared to the page limit.

---

> ### Author Response · Authors · 2022-08-02
> **Response to Reviewer GKqy**
>
> We are very grateful for the constructive comments and the time devoted to our work. Our responses to these comments are provided below.
>
> **[Q1. Running time.]** Great point. The running time of the proposed methods and the baselines are shown in Figures 11a and 11b in Appendix F.5. We observe that the proposed method has a longer running time when the number of variables is small, which may not be surprising because, different from the other one-stage imputation methods, our proposed method inherits the iterative optimization property of EM method, leading to a longer running time. Nevertheless, as soon as the number of variables gets very large (i.e., more than $640$ variables), our method runs faster than the other strong baselines (i.e., Gaussian-EM and MissForest imputations) since these methods also take much time for imputation. Moreover, the running time of the other baselines appears to increase more quickly w.r.t. the number of variables as compared to our proposed method. These observations indicate that our method appears to scale well, in addition to the improvement of structure learning performance observed in the experiments in Appendix F. We have included this discussion in Appendix G.2 of the revision.
>
> **[Q2. Convergence analysis.]** Thanks for pointing this out. Our method inherits similar convergence results to the EM algorithm. Specifically, the convergence conditions of our method depend on the optimization method since our graph solution space is totally countable. For complete data, our studied models are all identifiable. That is to say, the log-likelihood can also uniformly converge to the function, of which the maximum is unique and corresponds to the ground-truth SCM. With the ignorable missingness assumption, we can identify the ground-truth graph by equivalently maximizing the log-likelihood of the visible part of the observations.
>
> To ensure the increase of log-likelihood in each EM iteration, we need the Eq. (27), i.e., $\mathcal{Q}(\theta^{(t+1)},\theta^{(t)}) \geq \mathcal{Q}(\theta^{(t)},\theta^{(t)})$ hold. By taking exact search methods, which can search the total parameters space, this can be guaranteed Since $\theta^{(t+1)}=\arg \max_{\theta}\mathcal{Q}(\theta,\theta^{(t)})$ can be guaranteed. However, if we take gradient-based methods to solve the optimization, we can only reach the stationary points of the overall optimization problem. Since DAG constraint is non-convex, we cannot guarantee to find better $\theta^{(t+1)}$.
>
> Even though there is no strict theoretical guarantee for this method, experimental results provided in our paper still show the effectiveness of our MissDAG. We have updated this discussion in Appendix E to make it clear.
>
> **[Q3. Minor points.]** Thanks for the comments which help improve the presentation. We will fix the typos and make the notations clear in the revision according to your suggestions.

---

> > ### Comment · Reviewer_GKqy · 2022-08-10
> > **Happy with author response**
> >
> > I am happy with the authors' answers to my concerns, and with what I read in the rest of the discussion. Also because I do not consider myself an expert on the topic of this paper and the other reviewers are more positive, I have increased my score from 4 to 6.

---

> ### Author Response · Authors · 2022-08-06
> **Response to Reviewer GKqy**
>
> Dear Reviewer GKqy,
>
> We appreciate your comments and time! We have provided answers to your questions and revised the paper following your suggestions. Would you mind checking it and confirming if you have further questions?
>
> Authors

---

> ### Author Response · Authors · 2022-08-08
> **Response to Reviewer GKqy**
>
> Dear Reviewer GKqy,
>
> We have provided answers to your questions and revised the paper following your suggestions. Would you mind checking it and confirming if you have further questions? Because there are only two days left for us to answer your questions.
>
> Authors

---

> ### Author Response · Authors · 2022-08-09
> **Response to Reviewer GKqy**
>
> Dear Reviewer GKqy,
>
> Thanks for your great efforts in reviewing our paper. We have provided answers to your questions and revised the paper following your suggestions. Would you mind checking it and confirming if you have further questions? Because there is only one day left for us to answer your questions.
>
> Authors

---

### Meta-Review · Area_Chair_oBJf · 2022-08-28

**Recommendation:** Accept
**Confidence:** Certain

**Metareview:**


All reviewers were convinced by the scientific value and results about this paper and voted for acceptance.
The authors did a good job addressing thoroughly the various concerned raised.
As a result, two of the reviewers happily increased their score.
A common concern remaining is the dense writing. The reviewers underlined the need
for the authors to improve the clarity and presentation. Acceptance is recommended.

**Award:**

No

---

### Decision · Program_Chairs · 2022-09-14

Accept